# OPERA: Automatic Offline Policy Evaluation with Re-weighted Aggregates of Multiple Estimators

**Allen Nie**[1]    **Yash Chandak**[1]    **Christina J. Yuan**[2]
**Anirudhan Badrinath**[1]    **Yannis Flet-Berliac**[3]    **Emma Brunskill**[1]*
[1]Computer Science, Stanford University
[2]Computer Science, University of Texas, Austin
[3]Cohere

## Abstract

Offline policy evaluation (OPE) allows us to evaluate and estimate a new sequential decision-making policy's performance by leveraging historical interaction data collected from other policies. Evaluating a new policy online without a confident estimate of its performance can lead to costly, unsafe, or hazardous outcomes, especially in education and healthcare. Several OPE estimators have been proposed in the last decade, many of which have hyperparameters and require training. Unfortunately, choosing the best OPE algorithm for each task and domain is still unclear. In this paper, we propose a new algorithm that adaptively blends a set of OPE estimators given a dataset without relying on an explicit selection using a statistical procedure. We prove that our estimator is consistent and satisfies several desirable properties for policy evaluation. Additionally, we demonstrate that when compared to alternative approaches, our estimator can be used to select higher-performing policies in healthcare and robotics. Our work contributes to improving ease of use for a general-purpose, estimator-agnostic, off-policy evaluation framework for offline RL.

## 1   Introduction

Offline reinforcement learning (RL) involves learning better sequential decision policies from logged historical data, such as learning a personalized policy for math education software (Mandel et al., 2014; Ruan et al., 2024), providing treatment recommendations in the ICU (Komorowski et al., 2018; Luo et al., 2024) or learning new controllers for robotics(Kumar et al., 2020; Yu et al., 2020). Offline policy evaluation (OPE), in which the performance $J(\pi_e)$ of a new evaluation policy $\pi_e$ is estimated given historical data, is a common subroutine in offline RL for policy selection, and can be particularly important when deciding whether to deploy a new decision policy that might be unsafe or costly. Offline policy evaluation methods estimate the performance of an evaluation policy $\pi_e$ given data collected by a behavior policy $\pi_b$. There are many existing OPE algorithms, including those that create importance sampling-based estimators (IS) (Precup, 2000), value-based estimators (FQE) (Le et al., 2019), model-based estimators (Paduraru, 2013; Liu et al., 2018b; Dong et al., 2023), doubly robust estimators Jiang and Li (2016); Thomas and Brunskill (2016), and minimax-style estimators (Liu et al., 2018a; Nachum et al., 2019; Yang et al., 2020).

This raises an important practical question: given a set of different OPE methods, each producing a particular value estimate for an evaluation policy, what value estimate should be returned? A simple approach is to avoid the problem and pick only one OPE algorithm or look at the direction of a set of OPE algorithms' scores as a coarse agreement measure. Voloshin et al. (2021) offered heuristics based

---

[1]Emails: {anie,ychandak,ebrun}@cs.stanford.edu. [3]Work done while at Stanford.

38th Conference on Neural Information Processing Systems (NeurIPS 2024).

on high-level domain structure (e.g., horizon length, stochasticity, or partial observability), but this does not account for instance-specific information related to the offline dataset adn policies.

In this paper we seek to aggregate the results of a set of multiple off-policy RL estimators to produce a new estimand with low mean square error. This work is related to several streams of prior work: (1) multi-armed bandit and RL algorithms that combine two estimands to yield a more accurate estimate; (2) multi-armed bandit and RL algorithms that select a single estimand out of a set of estimands, and (3) stacked generalization / meta-learning / super learning methods in machine learning.

Research in (1) builds on doubly robust (DR) estimation in statistics to produce an estimand that combines important sampling and model-based methods (Jiang and Li, 2016; Gottesman et al., 2019; Farajtabar et al., 2018). Similarly, accounting for multiple steps, the MAGIC estimator blends between IS-based and value-based estimators within a trajectory (Thomas and Brunskill, 2016). The second line of work (2) does not combine scores but instead introduces an automatic estimator selection subroutine in the algorithm. However, such methods typically assume strong structural requirements on the input estimators. For example, Su et al. (2020); Tucker and Lee (2021) assume as input a nested set of OPE estimators, where the bias is known to strictly decrease across the set. Zhang and Jiang (2021) leveraged a set of Q-functions trained with fitted Q-evaluation (FQE) and cross-compare them in a tournament style until one Q-function emerged. None of these methods allow mix-and-match of different kinds of OPE estimators.

Our work is closest to a third line of more distant work, that of stacked generalization (Wolpert, 1992) / meta-learning and super learning across ensembles. There is a long history in statistics and supervised learning of combining multiple input classification or regression functions to produce a better meta-function. Perhaps surprisingly, there is little exploration of this idea to our knowledge in the context of RL or multi-armed bandits. The one exception we are aware of was for heterogeneous treatment effect estimation in a 2-action contextual bandit problem, where Nie and Wager (2021) utilized linear stacking to build a consensus treatment effect estimate using two input estimatands.

In this paper we introduce the meta-algorithm OPERA (**O**ffline **P**olicy **E**valuation with **R**e-weighted **A**ggregates of Multiple Estimators). Inspired by a linear weighted stack, OPERA combines multiple generic OPE estimates for RL in an ensemble to produce an aggregate estimate. Unlike in supervised learning where ground truth labels are available, in our setting a key choice is how to estimate the mean squared error of the resulting weighted ensemble. Under certain conditions, bootstrapping (Efron, 1992) can approximate finite sample bias and variance. We use bootstrapping to compute estimates of the mean squared error of different weightings of the underlying input estimators, which can then be optimized as a constrained convex problem. OPERA can be used with any input OPE estimands. We prove under mild conditions that OPERA produces an estimate that is consistent, and will be at least as accurate as any input estimand. We show on several common benchmark tasks that OPERA achieves more accurate offline policy evaluation than prior approaches, and we also provide a more detailed analysis of the accuracy of OPERA as a function of choices made for the meta-algorithm.

## 2 Related Work

**Offline policy evaluation**   Most commonly used offline policy estimators can be divided into a few categories depending on the algorithm. An important family of estimators focuses on using importance sampling (IS) and weighted importance sampling (WIS) to reweigh the reward from the behavior policy (Precup, 2000). These estimators are known to produce an unbiased estimate but have a high variance when the dataset size is small. For a fully observed Markov Decision Process (MDP), a model-free estimator, such as fitted Q evaluation (FQE), is proposed by Le et al. (2019), and one can also learn a model given the data to produce a model-based (MB) estimate (Pǎduraru, 2007; Fu et al., 2021; Gao et al., 2023). When the behavior policy's probability distribution over action is unknown, a minimax style optimization estimator (DualDICE) can jointly estimate the distribution ratio and the policy performance (Nachum et al., 2019). For a partial observable MDP (POMDP), many of these methods have been extended to account for unobserved confounding, such as minimax style estimation (Shi et al., 2022), value-based estimation (Tennenholtz et al., 2020; Nair and Jiang, 2021), uses sensitivity analysis to bound policy value (Kallus and Zhou, 2020; Namkoong et al., 2020; Zhang and Bareinboim, 2021), or learns useful representation over latent space (Chang et al., 2022).

**OPE with multiple estimators**   Choosing the right estimators has become an issue when there are many proposals even under the same task setup and assumptions. Voloshin et al. (2021) proposed

an empirical guide on estimator selection. One line of work tries to combine multiple estimators to produce a better estimate by leveraging the strengths of the underlying estimators, for example, (weighted) doubly robust (DR) method (Jiang and Li, 2016). For contextual bandit, Wang et al. (2017) proposed a switch estimator that interpolates between DM and DR estimates with an explicitly set hyperparameter. For sequential problems, MAGIC blends a model-based estimator and guided importance sampling estimator to produce a single score (Thomas and Brunskill, 2016). Another line of work tackles the many-estimator problem by reformulating multiple estimators as one estimator. Yang et al. (2020) reformulated a set of minimax estimators as a single estimator with different hyperparameter configurations. Yuan et al. (2021) constructed a spectrum of estimators where the endpoints are an IS estimator and a minimax estimator and proposed a hyperparameter to control the new estimator. This line of approaches does not leverage multiple estimators or solve the OPE selection problem because they recast the OPE selection problem as a hyperparameter selection problem. The last line of work provides an automatic selection algorithm that chooses one estimator from many, relying on an ordering of estimators (Tucker and Lee, 2021) or being able to compare the output (such as Q-values) directly Zhang and Jiang (2021).

**Bootstrapping for model selection**   Using bootstrap to estimate the mean-squared error for model selection was initially proposed by Hall (1990), for the application of kernel density estimation. The idea was subsequently used by others for density estimation (Delaigle and Gijbels, 2004), selecting sample fractions for tail index estimation (Danielsson et al., 2001), time-series forecasting (dos Santos and Franco, 2019) and other econometric applications (Marchetti et al., 2012). Similar ideas have been explored by Thomas et al. (2015) to construct a confidence interval for the estimator. We extend this idea to use bootstrapping to combine multiple OPE estimators to produce a single score.

# 3   Notation and Problem Setup

We define a stochastic Decision Process $M = \langle \mathcal{S}, A, T, r, \gamma \rangle$, where $\mathcal{S}$ is a set of states; $A$ is a set of actions; $T$ is the transition dynamics; $r$ is the reward function; and $\gamma \in (0, 1)$ is the discount factor. Let $D_n = \{\tau_i\}_{i=1}^n = \{s_i, a_i, s_i', r_i\}_{i=1}^n$ be the trajectories sampled from $\pi$ on $M$. We denote the true performance of a policy $\pi$ as its expected discounted return $J(\pi) = \mathbb{E}_{\tau \sim \rho_\pi}[G(\tau)]$ where $G(\tau) = \sum_{t=0}^\infty \gamma^t r_t$ and $\rho_\pi$ is the distribution of $\tau$ under policy $\pi$. In an off-policy policy evaluation problem, we take a dataset $D_n$, which can be collected by one or a group of policies which we refer to as the behavior policy $\pi_b$ on the decision process $M$. An OPE estimator takes in a policy $\pi_e$ and a dataset $D_n$ and returns an estimate of its performance, where we mark it as $\hat{V} : \Pi \times \mathcal{D} \to \mathbb{R}$. We focus on estimating the performance of a single policy $\pi$. We define the true performance of the policy $V^\pi = J(\pi)$, and multiple OPE estimates of its performance as $\hat{V}_i^\pi(D_n) = \hat{V}_i(\pi, D_n)$ for the $i$-th OPE's estimate.

# 4   OPERA

In this section, we consider combining results from multiple estimators $\{\hat{V}_i^\pi\}_{i=1}^k$ to obtain a better estimate for $V^\pi$. Towards this goal, given $\{\hat{V}_i^\pi\}_{i=1}^k$, we propose estimating a set of weights $\alpha_i^* \in \mathbb{R}$ such that $\bar{V}^\pi := \sum_{i=1}^k \alpha_i^* \hat{V}_i^\pi \in \mathbb{R}$ has the lowest mean squared error (MSE) towards estimating $V^\pi$. Formally, let $\hat{\mathcal{V}} \in \mathbb{R}^{k \times 1}$ be a vector whose elements correspond to values from different estimators, and let $\mathcal{V} \in \mathbb{R}^{k \times 1}$ correspond to a vector where each element is the same and corresponds to $V^\pi$. Let $\alpha^* \in \mathbb{R}^{k \times 1}$ be a vector with values of all $\alpha_i^*$'s and let $\alpha \in \mathbb{R}^{k \times 1}$ be an estimate of $\alpha^*$. For any estimator $\hat{V}_i^\pi$, the mean-squared error is denoted by,

$$\text{MSE}(\hat{V}_i^\pi) := \mathbb{E}_{D_n}\left[\left(\hat{V}_i^\pi(D_n) - V^\pi\right)^2\right] \in \mathbb{R}, \tag{1}$$

where we make $\hat{V}_i^\pi$ explicitly depend on $D_n$ to indicate that the expectation is over the random variables $\hat{V}_i^\pi$ which depend on the sampled data $D_n$. With this formulation, estimating $\alpha^*$ can be elicited as a solution to the following constrained optimization problem.

**Remark 1.** *Let $\sum_{i=1}^k \alpha_i = 1$, then*

$$\alpha^* \in \underset{\alpha \in \mathbb{R}^{k \times 1}}{\arg\min} \quad \alpha^\top A \alpha \quad , \text{where} \quad A := \mathbb{E}\left[\left(\hat{\mathcal{V}} - \mathcal{V}\right)\left(\hat{\mathcal{V}} - \mathcal{V}\right)^\top\right] \in \mathbb{R}^{k \times k}. \tag{2}$$

Using the fact that $\sum_{i=1}^{k} \alpha_i = 1$,

$$\mathrm{MSE}(\bar{V}^\pi) = \mathbb{E}\left[\left(\sum_{i=1}^{k}\alpha_i \hat{V}_i^\pi - V^\pi\right)^2\right] = \mathbb{E}\left[\left(\sum_{i=1}^{k}\alpha_i\left(\hat{V}_i^\pi - V^\pi\right)\right)^2\right]. \tag{3}$$

Now re-writing the equation above equation 3 in vector form,

$$\mathrm{MSE}(\bar{V}^\pi) = \mathbb{E}\left[\left(\left(\hat{\mathcal{V}} - \mathcal{V}\right)^\top \alpha\right)^2\right] = \mathbb{E}\left[\alpha^\top\left(\hat{\mathcal{V}} - \mathcal{V}\right)\left(\hat{\mathcal{V}} - \mathcal{V}\right)^\top \alpha\right]. \tag{4}$$

Finally, simplifying equation 4 further

$$\mathrm{MSE}(\bar{V}^\pi) = \alpha^\top \mathbb{E}\left[\left(\hat{\mathcal{V}} - \mathcal{V}\right)\left(\hat{\mathcal{V}} - \mathcal{V}\right)^\top\right]\alpha = \alpha^\top A \alpha. \tag{5}$$

Therefore, $\alpha$ that minimizes $\mathrm{MSE}(\bar{V}^\pi)$ is equivalent to $\alpha$ that minimizes $\alpha^\top A \alpha$.

It is worth highlighting that the optimization problem in Remark 1 is convex in $\alpha$ with linear constraint and thus can be solved by any off-the-shelf solvers (Diamond and Boyd, 2016).

**Estimating $A$:** An advantage of Remark 1 is that it provides the objective for estimating $\alpha^*$. Unfortunately, this objective depends on $A$, and thus on $V^\pi$, which is not available. Further, observe that $A$ can be decomposed as

$$A = \mathbb{E}\left[\left(\hat{\mathcal{V}} - \mathbb{E}\left[\hat{\mathcal{V}}\right]\right)\left(\hat{\mathcal{V}} - \mathbb{E}\left[\hat{\mathcal{V}}\right]\right)^\top\right] + \left[\mathbb{E}\left[\hat{\mathcal{V}} - \mathcal{V}\right]\left[\hat{\mathcal{V}} - \mathcal{V}\right]^\top\right]. \tag{6}$$

where the first term corresponds to co-variance between the estimators $\{V_i^\pi\}_{i=1}^{k}$ and the second term corresponds to the outer product between their biases. One potential approach for approximating $A$ could be to ignore biases. While this could resolve the issue of not requiring access to $\mathcal{V}$, ignoring bias can result in severe underestimation of $A$, especially in finite-sample settings or when function approximation is used. Further, even if we ignore the biases, it is not immediate how to compute the covariance of various OPE estimators, e.g., FQE.

We propose overcoming these challenges by constructing $\hat{A} \in \mathbb{R}^{k \times k}$, an estimate of $A \in \mathbb{R}^{k \times k}$, using a statistical bootstrapping procedure (Efron and Tibshirani, 1994). Subsequently, we will use the $\hat{A}$ as a plug-in replacement for $A$ to search for the values of $\alpha$ as discussed in Remark 1. There is a rich literature on using bootstrap to estimate bias (Efron, 1990; Efron and Tibshirani, 1994; Hong, 1999; Shi, 2012; Mikusheva, 2013) and variance (Chen, 2017b; Gamero et al., 1998; Shao, 1990; Ghosh et al., 1984; Li and Maddala, 1999) of an estimator that can be leveraged to estimate the terms in equation 6. Instead of estimating the bias and variance individually, we directly use the bootstrap MSE estimate (Chen, 2017a; Williams, 2010; Cao, 1993; Hall, 1990) to approximate $A$.

For bootstrap estimation to work, two key challenges need to be resolved. Even if provided with $V^\pi$, the regular bootstrap is not guaranteed to yield an MSE estimate which is asymptotic to the true MSE if the distribution has heavy tails (Ghosh et al., 1984). Furthermore, $V^\pi$ is unknown in the first place. To address these challenges we follow the work by Hall (1990), where the first issue is resolved by using sub-sampling based bootstrap resamples of size $n_1 < n$, where $n_1$ is of a smaller order than $n$. Therefore, we draw data $D_{n_1}^* = \{\tau_1^*, ..., \tau_{n_1}^*\}$ from $D_n = \{\tau_1, ..., \tau_n\}$ with replacement. To resolve the second issue, we leverage the MSE estimate by Hall (1990), and approximate equation 1 using

$$\widehat{\mathrm{MSE}}(\hat{V}_i^\pi) := \mathbb{E}_{D_{n_1}^*}\left[\left(\hat{V}_i^\pi(D_{n_1}^*) - \hat{V}_i^\pi\right)^2 \Big| D_n\right]. \in \mathbb{R} \tag{7}$$

Building upon this direction, we propose using the following estimator $\hat{A}$ for $A$,

$$\hat{A} := \mathbb{E}_{D_{n_1}^*}\left[\left(\hat{\mathcal{V}}(D_{n_1}^*) - \hat{\mathcal{V}}\right)\left(\hat{\mathcal{V}}(D_{n_1}^*) - \hat{\mathcal{V}}\right)^\top \Big| D_n\right], \in \mathbb{R}^{k \times k} \tag{8}$$

and we substitute $\hat{\alpha}$ for $A$ in equation 2 to obtain the weights for combining estimates $\{\hat{V}_i^\pi\}_{i=1}^{k}$

$$\hat{V}^\pi := \sum_{i=1}^{k} \hat{\alpha}_i \hat{V}_i^\pi \in \mathbb{R} \quad \text{where,} \quad \hat{\alpha} \in \underset{\alpha \in \mathbb{R}^{k \times 1}}{\arg\min} \ \alpha^\top \hat{A} \alpha \in \mathbb{R}^{k \times 1}. \tag{9}$$

There are two key advantages of the proposed procedure: (1) Bias: it does not require access to the ground truth performance estimates $\theta^*$. In the supervised learning setting, a held-out/validation set can provide a way to infer approximation error, However, for the OPE setting there is no such held-out dataset that can be used to obtain reliable estimates of the ground truth performance. (2) Variance: Depending on the choice of the estimator (e.g., FQE), it might not be possible to have a closed-form estimate of the variance, especially when using rich function approximators. Using statistical bootstrapping, OPERA mitigates both these issues and thus is particularly suitable for off-policy evaluation.

**Estimating $\alpha^*$:**  We now consider how error in estimating the optimal weight coefficient $\alpha^*$ affects the MSE of the resulting estimator $\hat{\bar{V}}^\pi$. Without loss of generality, we consider $|\hat{V}_i^\pi| \leq 1$, since we can trivially normalize each estimator's output by $|V_{\max}|$. We now prove that under the mild assumption that the error in the estimated $\hat\alpha$ can be bounded as some function of the dataset size, that we can bound the mean squared error of the resulting value estimate:

**Theorem 1** (Finite Sample Analysis). *Assume given $n$ samples in dataset $D$, and let $\Delta_c :=$* $\mathbb{E}_{D_n}\left[\left(\bar{V}^\pi - V^\pi\right)_i^2\right]$, *there exists a $\lambda > 0$ such that*

$$\forall i, \quad \mathbb{E}_{D_n}[|\hat\alpha_i - \alpha_i^*|] \leq n^{-\lambda}, \tag{10}$$

$$\mathrm{MSE}(\hat{\bar{V}}^\pi) \leq \frac{k^2}{n^{2\lambda}} + \Delta_c. \tag{11}$$

The error of OPERA is divided into two terms. First note that $\Delta_c$ is the approximation error: the difference between the true estimate of the policy performance $V^\pi$ and the best estimand OPERA can yield when using the optimal (unknown) $\alpha^*$. If $V^\pi$ can be expressed as a linear combination of the input OPE estimands $\hat\theta_i$, then there is zero approximation error and $\Delta_c = 0$. The second term in the bound comes from the estimation error due to estimating $\alpha^*$– this arises from the bootstrapping process used for estimating $A$ in equation 8. For this second term we compute an upper bound using a Cauchy-Schwartz inequality. This term decreases as the dataset size $n$ increases. The resulting error depends on the rate at which the estimated $\hat\alpha$ converges to the true $\alpha$ as a function of the dataset size. For example, if $\lambda = 0.5$, (a $n^{-.5}$ rate), the MSE will converge at a $n^{-1}$ rate in the first term, and if $\lambda = 0.25$ (a $n^{-.25}$ rate) the MSE will converge at a $n^{-0.5}$ rate in the first term. We provide the full proof in Appendix A.4.

We show a full practical implementation of OPERA in Algorithm 1, where we demonstrate how to efficiently construct $\hat A$ and compute $\hat\alpha$.

### 4.1 Properties of OPERA

For $\hat A$ obtained from the bootstrap procedure in equation 8 to be an asymptotically accurate estimate of $A$, (a) a consistent estimator of $\mathcal{V}$ is required, and (b) the estimators $\hat{\mathcal{V}}$ need to be smooth. We discuss these points in more detail in Appendix A.3. In the following, we theoretically establish the properties of OPERA on performance improvement and consistency. We also demonstrate how OPERA allows us to interpret each estimator's quality. Further, in Section 6, we empirically study the effectiveness of OPERA even when we do not have any consistent base estimators, or $\hat{V}_i^\pi$ is constructed using deep neural networks.

**Performance Improvement**  It would be ideal that the combined estimator $\hat{\bar{V}}^\pi$ does not perform worse than any of the base estimators $\{\hat{V}_i^\pi\}_{i=1}^n$. As OPERA optimizes for the MSE, we can directly obtain the following desired result.

**Theorem 2** (Performance improvement). *If $\hat\alpha = \alpha^*$, $\forall i \in \{1,...,k\}$, $\quad MSE(\hat{\bar{V}}^\pi) \leq MSE(\hat{V}_i^\pi)$.*

However, observe that due to bootstrap approximation, $\hat A$ may not be equal to $A$, and thus $\hat\alpha$ may not be equal to $\alpha^*$. Nonetheless, as we will illustrate in Section 6, even in the non-idealized setting OPERA can often achieve MSE better than any of the base estimators $\{V_i^\pi\}_{i=1}^n$.

**Consistency**  Some prior works that deal with multiple OPE estimators assume that there is at least one known consistent estimator (Thomas and Brunskill, 2016). Under a similar assumption that $\exists \hat{V}_i^\pi : \hat{V}_i^\pi \xrightarrow{p} J(\pi)$, OPERA can be made to fall back to the consistent estimator after a large $n$, such that $\hat{\bar{V}}^\pi$ is also consistent, i.e., $\hat{\bar{V}}^\pi \xrightarrow{p} J(\pi)$. Naturally, as $\bar{V}^\pi$ is a weighted combination

of the base estimators $\{\hat{V}_i^\pi\}_{i=1}^k$, if *all* the base estimators provide unreliable estimates, even in the limit of infinite data, then there is not much that can be achieved by weighted combinations of these unreliable estimators.

**Interpretability** With a linear weighted formulation for $\bar{V}^\pi$, OPERA allows for the inspection of the assigned weights to which give further insights into the procedure. In Figure 1 we provide a synthetic example to illustrate the impact of bias and variance of the input estimators on the values of $\alpha$.

Consider a case where there are two OPE estimators ($\hat{V}_1^\pi$ and $\hat{V}_2^\pi$) and two corresponding weights $\alpha_1$ and $\alpha_2$). Let the true unknown quantity be $V^\pi = 0$. As we can see below, when both estimators have low bias, but one has higher variance, OPERA assigns a higher magnitude of $\alpha_i$ for $V_i^\pi$ with a lower variance (Figure1, left). When both estimators have similar variance, and their biases have *opposite* signs with similar magnitude, then $\alpha_2 \approx \alpha_1$ (Figure1, middle left).

Interestingly, unlike related prior work (Thomas and Brunskill, 2016), our optimization procedure in equation 2 does not require $\alpha_i \geq 0$. Therefore the resulting estimator $\bar{V}^\pi$ may assign negative weights for some of the estimators.

---

**Algorithm 1:** OPERA with Bootstrap

**Input:** offline RL data $D_n$; evaluation policy $\pi$; a set of OPE estimators $[\text{OPE}_1, \text{OPE}_2, ..., \text{OPE}_k]$; number of bootstrap $B$; a subsample coefficient $\eta \in [0, 1]$.

**Output:** estimated $\pi$ performance $s_{\text{OPERA}}$

**for** $i \leftarrow 1...K$ **do**
  $s_i^* = \text{OPE}_i(D_n)$
  $\tilde{s}_i = \emptyset$
  **for** $j \leftarrow 1...B$ **do**
    $n_1 = |\mathcal{D}|^\eta$
    $\tilde{\mathcal{D}}_j \leftarrow \text{Bootstrap}(D_n, n_1)$
    $\tilde{s}_i = \tilde{s}_i \cup \text{OPE}_i(\tilde{D}_j)$
  **end**
**end**
$\tilde{M} \leftarrow [\tilde{s}_1, \tilde{s}_2, ..., \tilde{s}_k] \in \mathbb{R}^{K \times B}$
$M \leftarrow [s_1^*, s_2^*, ..., s_k^*] \in \mathbb{R}^{K \times 1}$
$\delta \leftarrow [(\tilde{s}_1 - s_1^*, \tilde{s}_2 - s_2^*, ..., \tilde{s}_k - s_k^*] \in \mathbb{R}^{K \times B}$
$A \leftarrow \frac{1}{B} \frac{n_1}{n} \delta \delta^\top \in \mathbb{R}^{K \times K}$
$\alpha = \arg\min_\alpha \alpha A \alpha^\top \quad s.t. \sum \alpha = 1$
$s_{\text{OPERA}} = \alpha^\top M$
**return** $s_{\text{OPERA}}$

---

This can be observed for the case when the *sign* of the $\text{Bias}(\hat{V}_1^\pi)$ and $\text{Bias}(\hat{V}_1^\pi)$ are the same. In such a case, using a positive and a negative weight can help cancel out the biases of the base estimators, as observed in Figure1 (middle right). When one estimator has no bias and the other has no variance, $\alpha$ values are inversely proportional to their contributions towards the MSE (Figure1, right).

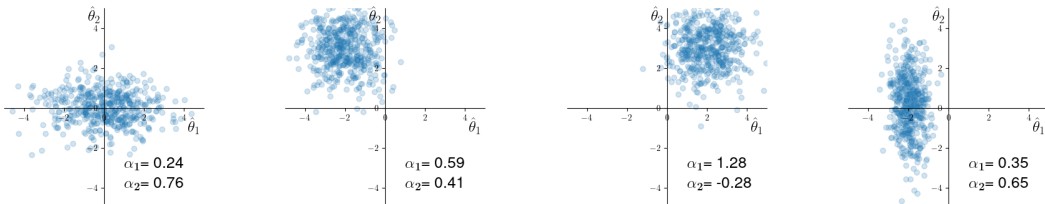

Figure 1: Interpreting weights for different estimators. X-axis shows the value of $\hat{V}_1^\pi$ and Y-axis shows the value of $\hat{V}_2^\pi$.

# 5 Experiment

We now evaluate OPERA on a number of domains commonly used for offline policy evaluation. Experimental details, when omitted, are presented in the appendix.

## 5.1 Task/Domains

**Contextual Bandit**. We validate the performance of OPERA on the synthetic bandit domain with a 10-dimensional feature space proposed in SLOPE (Su et al., 2020). This domain illustrates how OPERA compares to an estimator-selection algorithm (SLOPE) that assumes a special structure between the estimators. The true reward is a non-linear neural network function. The reward estimators are parametrized by kernels and the bandwidths are the main hyperparameters. As in their paper, we ran 180 configurations of this simulated environment with different parametrization of the environment. Each configuration is replicated 30 times.

**Sepsis**. This domain is based on a simulator that allows us to model learning treatment options for sepsis patients in ICU (Oberst and Sontag, 2019). There are 8 actions and a +1/-1/0 reward at the

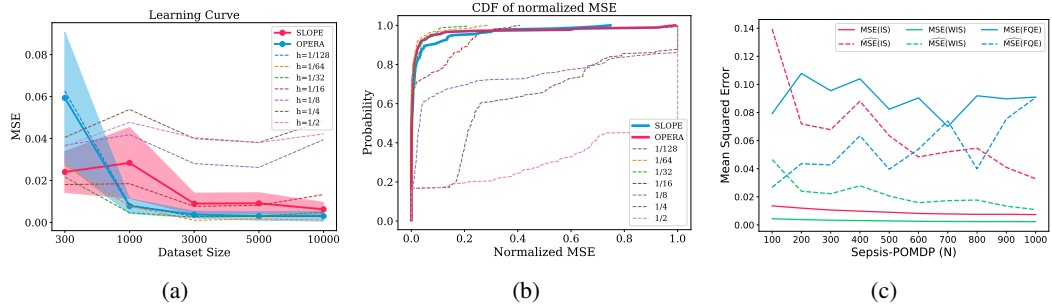

Figure 2: Left: Results for contextual bandits. (a) MSE of estimators when the dataset size grows. (b) CDF of normalized MSE across 180 conditions by the worst MSE of that condition. Better methods lie in the top-left quadrant. Right: (c) For an MDP domain (Sepsis), we show that as dataset sizes increase, our bootstrap estimation of MSE approaches true MSE for each OPE estimator.

episode end. We experiment with two settings: Sepsis-MDP and Sepsis-POMDP, where some crucial states have been masked. We evaluate 7 different policies: an optimal policy and 6 noised suboptimal policies, which we obtain by adding uniform noise to the optimal policy.

**Graph**. Voloshin et al. (2019) introduced a ToyGraph environment with a horizon length T and an absorbing state $x_{abs} = 2T$. Rewards can be deterministic or stochastic, with +1 for odd states, -1 for even states plus one based on the penultimate state. We evaluate the considered methods on a short horizon H=4, varying stochasticity of the reward and transitions, and MDP/POMDP settings.

**D4RL-Gym**. D4RL (Fu et al., 2020) is an offline RL standardized benchmark designed and commonly used to evaluate the progress of offline RL algorithms. We use 6 datasets (200k samples each) from three Gym environments: Hopper, HalfCheetah, and Walker2d. We use two datasets from each: the medium-replay dataset, which consists of samples from the experience replay buffer, and the medium dataset, which consists of samples collected by the medium-quality policy. We use conservative Q-learning (CQL) (Kumar et al., 2020), implicit Q-learning (IQL) (Kostrikov et al., 2021), and TD3 (Fujimoto et al., 2018). We train 6 policies from these three algorithms with 2 different hyperparameters for the neural network. We selected 2 FQE hyperparameters for each task and picked 2 checkpoints (one early, one late) to obtain 4 estimators to build the OPE ensemble.

### 5.2 Baseline Ensemble OPE Methods

We compare to using single OPE estimators as well as two new baseline algorithms that combine OPE estimates together. **AvgOPE**: We can compute a simple average estimator that just outputs the average of all underlying OPE estimates. If an estimator in the ensemble outputs an arbitrarily bad value, this estimator has no implicit mechanism to ignore such an adversarial estimator. **BestOPE**: We select the OPE estimator that has the smallest estimated MSE. This estimator can be better than AvgOPE as it can ignore bad estimators. In addition, in different domains, we compare to other OPE strategies such as **BVFT** (Batch Value Function Tournament): making pariwise comparisons between different Q-function estimators with the BVFT-loss (Xie and Jiang, 2021; Zhang and Jiang, 2021). **SLOPE**: an estimator selection method that based on Lepski's method, assuming the estimators forming an order of decreasing variance and increasing bias (Yuan et al., 2021). **DR** (Doubly Robust): a semi-parametric estimator that combines the **IS** estimator and **FQE** estimator to have an unbiased

Table 1: We report the Mean-Squared Error (MSE) for the Sepsis domain. Each number is averaged across 20 trials. We underscore the estimator that has the lowest MSE in the ensemble.

| Sepsis | N | OPERA | IS | WIS | FQE |
|--------|------|--------|--------|--------|--------|
| MDP | 200 | **0.2205** | 0.2753 | 0.2998 | 0.2448 |
| MDP | 1000 | **0.1705** | 0.1720 | 0.2948 | 0.2995 |
| POMDP | 200 | **0.2750** | 0.2804 | 0.2850 | 0.3931 |
| POMDP | 1000 | **0.2749** | 0.2799 | 0.3092 | 0.4078 |

low variance estimator (Jiang and Li, 2016; Gottesman et al., 2019; Farajtabar et al., 2018). All of these methods place explicit constraints on the type of OPE estimator to include.

Table 2: Root Mean-Squared Error (RMSE) of different OPE algorithms across D4RL tasks.

| Env/Dataset | Multi-OPE Estimator | | | | | Single OPE Estimator | |
| --- | --- | --- | --- | --- | --- | --- | --- |
| | OPERA | BestOPE | AvgOPE | BVFT | DR | Dual-DICE | MB |
| **Hopper** | | | | | | | |
| medium-replay | **13.0** | 15.5 | 60.7 | 61.2 | 112.7 | 1565.2 | 298.7 |
| medium | **8.5** | 12.5 | 120.8 | 16.4 | 16.5 | 368.58 | 269.7 |
| **HalfCheetah** | | | | | | | |
| medium-replay | **46.0** | 65.0 | 218.6 | 140.2 | 119.5 | 567.9 | 750.9 |
| medium | **100.5** | 111.8 | 262.1 | 166.6 | 145.2 | 3450.0 | 589.9 |
| **Walker2d** | | | | | | | |
| medium-replay | **138.3** | 167.4 | 187.2 | 221.5 | 155.3 | 2124.3 | 316.8 |
| medium | **149.0** | 183.8 | 859.4 | 264.1 | 232.1 | 1756.4 | 1269.3 |

## 5.3 Results

**Contextual Bandit**   We report the result in Figure 2. Figure 2a shows that as the dataset size grows, the bootstrapping procedure employed by OPERA can quickly estimate the performance each estimator and compute a weighted score that is better than a single estimator. In the ultra-small data regime, OPERA is worse than single-estimator selection style algorithms, mainly because OPERA does not explicitly reject estimators. We can add an additional procedure to reject bad estimators and then combine the rest with OPERA, using a rejection algorithm by Lee et al. (2022).

**Sepsis**   We report the results in Table 1. In this domain, OPERA is able to produce an estimate, on average, across many policies with different degrees of optimality, that matches and exceeds the best estimator in the ensemble. Even though in three out of four tasks, OPERA MSE is close to the MSE of the best estimator in the ensemble, in the MDP (N=200) setup, OPERA is able to get a significantly lower MSE than any of the estimators in the ensemble, suggesting a future direction of carefully choosing a set of weak estimators to put in the ensemble to obtain a strong estimator.

**Graph**   We report the graph domain result in Appendix A.9 and in Table 5. We find a similar result to the Sepsis domain. OPERA is able to outperform AvgOPE and BestOPE in different setups.

**D4RL**   We report the results in Table 2. We choose this domain because, in continuous control tasks, the horizon is often very long. Many OPE estimators that rely on short-horizon or discrete actions will not be able to extend to this domain. A popular OPE choice is FQE with function approximation, but it is difficult to determine hyperparameters like early stopping, network architecture, and learning rate. We can see that even though FQE used in D4RL is not a consistent estimator and does not satisfy OPERA's theoretical assumption, we are still able to combine the estimations to reach an aggregate estimate with lower MSE.

## 6   Discussion: Different MSE Estimation Strategies

### 6.1   Estimating MSE with MAGIC

Part of our algorithm implicitly involves estimating the MSE of each OPE estimator. In our algorithm we do this using bootstrapping but other alternatives are possible. For example, prior work by (Thomas and Brunskill, 2016) provided a way to estimate the bias and variance of an OPE estimator are computed through per-trajectory OPE scores and used this as part of their MAGIC estimator. However, this method cannot estimate the MSE of self-normalizing estimators (such as WIS) or minimax-style estimators (such as any estimator in the DICE family (Yang et al., 2020)). We denote this estimator as $\widehat{\text{MSE}}_{\text{MAGIC}}(V^\pi)$ and now explore how our approach of using boostrapping compares to this method in an illustrative setting.

In particular, we consider estimating the MSE of the FQE and IS estimands on the Sepsis-POMDP and Sepsis-MDP domains. MAGIC estimates the bias of an OPE as the distance between the OPE

Table 3: We compare two styles of MSE estimations and how well they can estimate the true MSE of each estimator. We report averaged results over 10 trials, with N=200.

| | Sepsis-POMDP | | | Sepsis-MDP | | |
|---|---|---|---|---|---|---|
| | $\text{MSE}(V^\pi)$ | $\widehat{\text{MSE}}_{\text{MAGIC}}(V^\pi)$ | $\widehat{\text{MSE}}(V^\pi)$ | $\text{MSE}(V^\pi)$ | $\widehat{\text{MSE}}_{\text{MAGIC}}(V^\pi)$ | $\widehat{\text{MSE}}(V^\pi)$ |
| IS | 0.0161 | 0.0281 | **0.0088** | 0.3445 | **0.0485** | 0.0056 |
| FQE | 0.0979 | 0.4953 | **0.0163** | 0.0077 | 0.0771 | **0.0011** |

Table 4: We report the Mean-Squared Error (MSE) for the Sepsis domain. We additionally present two variants of OPERA where we experimented with different MSE estimation strategies.

| Sepsis | N | OPERA | OPERA-IS | OPERA-MAGIC | IS | WIS | FQE |
|---|---|---|---|---|---|---|---|
| MDP | 200 | 0.2205 | **0.2181** | 0.2657 | 0.2753 | 0.2998 | 0.2448 |
| MDP | 1000 | **0.1705** | 0.1779 | 0.1848 | 0.1720 | 0.2948 | 0.2995 |
| POMDP | 200 | **0.2750** | 0.2768 | 0.2827 | 0.2804 | 0.2850 | 0.3931 |
| POMDP | 1000 | 0.2749 | **0.2720** | 0.2802 | 0.2799 | 0.3092 | 0.4078 |

value and the closest upper or lower bound of a weighted importance sampling (WIS) policy estimate. We use a percentile bootstrap to construct a 50% confidence interval CI around WIS.

Our bootstrap $\widehat{\text{MSE}}(V^\pi)$ procedure is able to provide a consistently more accurate estimate of the true MSE of the FQE estimate compared to $\widehat{\text{MSE}}_{\text{MAGIC}}(V^\pi)$ and a comparable or better one for the IS estimate (see Table 3). We suspect that this is due to MAGIC's unique way of computing bias. Specifically, MAGIC computes bias by comparing two estimates (in this case, FQE and the uppper/lower bounds on WIS) which may significantly misestimate the bias in some situations.

### 6.2 Variants of OPERA with Different Strategies

We now explore two alternative strategies to estimate the MSE of each estimator. The first strategy is, instead of using the estimator's own score as the centering variable $\hat{\mathcal{V}}$, we use a consistent and unbiased estimator's score as $\hat{\mathcal{V}}$. We call this OPERA-IS. Another strategy is to use the idea from (Thomas and Brunskill, 2016)'s MAGIC algorithm, where the bias estimate of each estimator compares the estimand to the upper or lower confidence bound of a weighted importance sampling estimator, as above. We call this OPERA-MAGIC. These are two new variants of our OPERA algorithm that will may lead to learning different $\hat{\alpha}$ weights and producing different linearly stacked estimates. We use these two new methods, and compute the true MSE of the resulting stacked estimate, compared to OPERA and other baseline estimates. We use the Sepsis domains to illustrate the results and use as input IS, WIS and FQE OPE estimates.

The true MSE of the resulting estimates are presented in Table 4. While using an unbiased consistent estimator as the centering variable can help further improve OPERA's estimate, sometimes it also hurts the performance (MDP N=1000 setting). OPERA-MAGIC however almost always performs worse than the best estimator in the ensemble. This suggests that when combining OPE scores this bound on the bias, which will provide a distorted estimate of the estimator bias especially in low data regimes, can lead to learning less effective weightings of the input OPE estimands. OPERA remains a solid option across all settings presented in the table.

## 7 Conclusion

We propose a novel offline policy evaluation algorithm, OPERA, that leverages ideas from stack generalization to combine many OPE estimators to produce a single estimate that achieves a lower MSE. Though such stacked generalization / meta-learning has been frequently used to create better estimates from ensembles of input methods in supervised learning, to our knowledge this is the first time it has been explored in offline reinforcement learning. One challenge is that unlike in supervised learning, we do not have ground truth labels for offline policy learning. OPERA uses bootstrapping to estimate the MSE for each OPE estimator in order to find a set of weights to blend each OPE's estimate. We provide a finite sample analysis of OPERA's performance under mild assumptions, and demonstrate that OPERA provides notably more accurate offline policy evaluation

estimates compared to prior methods in benchmark bandit tasks and offline RL tasks, including a Sepsis simulator and the D4RL settings. There are many interesting directions for future work, including using more complicated meta-aggregators.

## Acknowledgments and Disclosure of Funding

The research reported in this paper was supported in part by a Stanford HAI Hoffman-Yee grant and an NSF #2112926 grant. We thank Sanath Kumar Krishnamurthy, Omer Gottesman, Yi Su, and Adith Swaminathan for the discussions.

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

# A Appendix

## A.1 OPERA Diagram

We describe the OPERA framework as a two-stage process in Figure 3. The first stage involves using black-box statistical methods (such as Bootstrap) to estimate the quality of each OPE estimator. The information is then used to estimate a weight $\alpha$ through a convex optimization objective in Eq 9. At stage two, we use the learned $\alpha$ to combine each estimator's score to output the OPERA score.

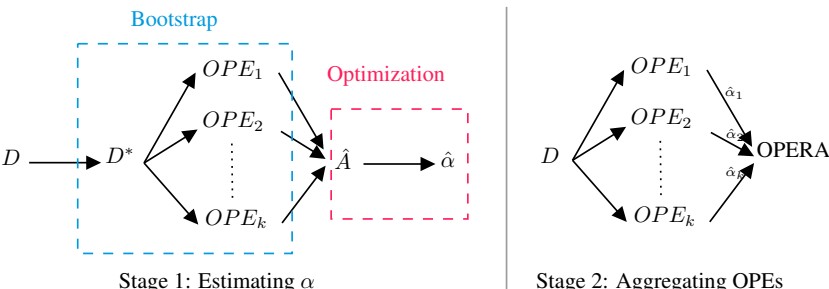

Figure 3: OPERA framework as a two-stage process.

## A.2 Notation Table

We provide a list of important notations used in this paper.

| Notation | Description |
|---|---|
| $n$ | We use $n$, $n_1$ to refer to the number of trajectories in the dataset. |
| $k$ | We use $k$ to refer to the number of estimators in the ensemble. |
| $D_n$ | $D_n = \{\tau_i\}_{i=1}^n = \{s_i, a_i, s_i', r_i\}_{i=1}^n$ be the trajectories sampled from $\pi$ on an MDP. |
| $D_{n_1}^*$ | A bootstrapped sample from $D_n$ with $n_1$ data points, $n_1 < n$. |
| $A$ | The cross-estimator bias variance matrix of $k$ estimators. $A \in \mathbb{R}^{k \times k}$ |
| $\hat{A}$ | The cross-estimator bias variance matrix of $k$ estimators using $D_{n_1}^*$. $\hat{A} \in \mathbb{R}^{k \times k}$ |
| $\alpha$ | Coefficient for $k$ estimators. $\alpha \in \mathbb{R}^{k \times 1}$ |
| $\alpha^*$ | Coefficient for $k$ estimators for solving $\alpha^* \in \arg\min_{\alpha \in \mathbb{R}^{k \times 1}} \alpha^\top A \alpha$. |
| $\hat{\alpha}$ | Coefficient for $k$ estimators for solving $\hat{\alpha} \in \arg\min_{\alpha \in \mathbb{R}^{k \times 1}} \alpha^\top \hat{A} \alpha$. |
| $V^\pi$ | True performance of the policy $\pi$, $V^\pi = J(\pi)$. |
| $\hat{V}_i^\pi$ | Let $i$-th OPE estimate of $\pi$'s performance be $\hat{V}_i^\pi(D_n) = \hat{V}_i(\pi, D_n)$. We use $\hat{V}_i^\pi = \hat{V}_i^\pi(D_n)$ where $D_n$ is the full offline dataset. |
| $\bar{V}^\pi(\alpha)$ | Weighted average of $k$ estimators with given $\alpha$: $\bar{V}^\pi(\alpha) := \sum_{i=1}^k \alpha_i \hat{V}_i^\pi \in \mathbb{R}$ |
| $\hat{\bar{V}}^\pi$ | OPERA's estimated performance of $\pi$. $\hat{\bar{V}}^\pi = \bar{V}^\pi(\hat{\alpha})$. |
| $\bar{V}^\pi$ | A shorthand notation for $\bar{V}^\pi = \bar{V}^\pi(\alpha^*)$. |
| $\mathrm{MSE}(\hat{V}_i^\pi)$ | The mean-squared error of the $i$-th OPE estimator. Based on how we constructed $A$, we have $\mathrm{MSE}(\hat{V}_i^\pi) = A_{i,i}$. |
| $\mathrm{MSE}(\hat{\bar{V}}^\pi)$ | The mean-squared error of $\hat{\bar{V}}^\pi$ (OPERA score). $\mathrm{MSE}(\hat{\bar{V}}^\pi) \leq \Delta_\alpha + \Delta_c$. |
| $\widehat{\mathrm{MSE}}(\hat{V}_i^\pi)$ | The estimated mean-squared error of the $i$-th estimator using bootstrapped dataset $D_{n_1}^*$. Based on how we constructed $\hat{A}$, we have $\widehat{\mathrm{MSE}}(\hat{V}_i^\pi) = \hat{A}_{i,i}$. |
| $\Delta_c$ | The change in mean-squared error of OPERA, $\hat{\bar{V}}^\pi$, from using dataset $D_n$. $\Delta_c := \mathbb{E}_{D_n}\left[\left(\bar{V}^\pi - V^\pi\right)^2\right]$. |
| $\Delta_\alpha$ | The change in mean-squared error of OPERA, $\hat{\bar{V}}^\pi$, from using sub-optimal $\alpha$. $\Delta_\alpha := \mathbb{E}_{D_n}\left[(\hat{\bar{V}}^\pi - \bar{V}^\pi)^2\right]$. |

### A.3 Bootstrap Convergence

In this section, we provide a high-level discussion of the bootstrap procedure and its asymptotic validity. We refer the readers to the works by (Cao, 1993; Hall, 1990) for a more fine-grained analysis and convergence rates when estimating MSE using statistical bootstrap. Individual treatment of bias (Efron, 1990; Efron and Tibshirani, 1994; Hong, 1999; Shi, 2012; Mikusheva, 2013) and variance (Chen, 2017b; Gamero et al., 1998; Shao, 1990; Ghosh et al., 1984; Li and Maddala, 1999) can also be found.

In the following, we will discuss the consistency of $\hat{A}$ estimated using bootstrap,

$$\hat{A}_{i.j} - A_{i,j} \xrightarrow{a.s.} 0. \tag{12}$$

Towards this goal, we will consider the following conditions imposed on the set of the base estimators $\{\hat{V}_i^\pi\}_{i=1}^k$,

- $\forall i, \quad \hat{V}_i^\pi$ is uniformly bounded.
- $\forall i, \quad \hat{V}_i^\pi \xrightarrow{a.s.} c_i$.
- $\forall i, \quad \hat{V}_i^\pi$ is smooth with respect to data distribution.
- $\exists \hat{V}_k^\pi : \hat{V}_k^\pi \xrightarrow{a.s.} c_k = V^\pi$.

Recall from equation 2,

$$A_{i,j} = \mathbb{E}\left[\left(\hat{V}_i^\pi - V^\pi\right)\left(\hat{V}_j^\pi - V^\pi\right)\right] \tag{13}$$

$$= \mathbb{E}\left[\left(\hat{V}_i^\pi - \mathbb{E}[\hat{V}_i^\pi] + \mathbb{E}[\hat{V}_i^\pi] - V^\pi\right)\right. \tag{14}$$

$$\left.\left(\hat{V}_j^\pi - \mathbb{E}[\hat{V}_j^\pi] + \mathbb{E}[\hat{V}_j^\pi] - V^\pi\right)\right] \tag{15}$$

$$= \mathbb{E}\left[\left(\hat{V}_i^\pi - \mathbb{E}[\hat{V}_i^\pi]\right)\left(\hat{V}_j^\pi - \mathbb{E}[\hat{V}_j^\pi]\right)\right] \tag{16}$$

$$+ \mathbb{E}\left[\left(\mathbb{E}[\hat{V}_i^\pi] - V^\pi\right)\left(\mathbb{E}[\hat{V}_j^\pi] - V^\pi\right)\right]. \tag{17}$$

Let $X_n := \left(\hat{V}_i^\pi - \mathbb{E}[\hat{V}_i^\pi]\right)$ and $Y_n := \left(\hat{V}_j^\pi - \mathbb{E}[\hat{V}_j^\pi]\right)$. As $\hat{V}_i^\pi \xrightarrow{a.s.} c_i$ and $\hat{V}_i^\pi$ is uniformly bounded, using (Thomas and Brunskill, 2016, Lemma 2), we have $\mathbb{E}[\hat{V}_i^\pi] \xrightarrow{a.s.} c_i$. Similarly, we have $\mathbb{E}[\hat{V}_j^\pi] \xrightarrow{a.s.} c_j$ as $\hat{V}_j^\pi \xrightarrow{a.s.} c_j$. Then using continuous mapping theorem,

$$X_n Y_n \xrightarrow{a.s.} (c_i - c_i)(c_j - c_j) = 0. \tag{18}$$

Now using (Thomas and Brunskill, 2016, Lemma 2),

$$\mathbb{E}\left[\left(\hat{V}_i^\pi - \mathbb{E}[\hat{V}_i^\pi]\right)\left(\hat{V}_j^\pi - \mathbb{E}[\hat{V}_j^\pi]\right)\right] = \mathbb{E}[X_n Y_n] \xrightarrow{a.s.} 0. \tag{19}$$

Similarly,

$$\left(\mathbb{E}[\hat{V}_i^\pi] - V^\pi\right)\left(\mathbb{E}[\hat{V}_j^\pi] - V^\pi\right) \xrightarrow{a.s.} (c_i - V^\pi)(c_j - V^\pi) \tag{20}$$

Therefore, using equation 19 and equation 20,

$$A_{i,j} \xrightarrow{a.s.} 0 + (c_i - V^\pi)(c_j - V^\pi). \tag{21}$$

Now we consider the asymptotic property of the bootstrap estimate $\hat{A}$ of $A$.

$$\hat{A}_{i,j} = \mathbb{E}_{D_{n_1}^*|D_n}\left[\left(\hat{V}_i^\pi(D_{n_1}^*) - \hat{V}_k^\pi\right)\left(\hat{V}_j^\pi(D_{n_1}^*) - \hat{V}_k^\pi\right)\right] \tag{22}$$

where $\hat{V}_k^\pi$ is known to be a consistent estimator, i.e., $\hat{V}_k^\pi \xrightarrow{a.s.} V^\pi$. Here, $\hat{V}_k^\pi$ could be the WIS or IS or doubly-robust estimators that are known to provide consistent estimates of $V^\pi = J(\pi)$. For brevity, we drop the conditional notation on the subscript, and write equation 22 as,

$$\hat{A}_{i,j} = \mathbb{E}_{D_{n_1}^*}\left[\left(\hat{V}_i^\pi(D_{n_1}^*) - \hat{V}_k^\pi\right)\left(\hat{V}_j^\pi(D_{n_1}^*) - \hat{V}_k^\pi\right)\right] \tag{23}$$

Simplifying equation 23,

$$\hat{A}_{i,j} = \mathbb{E}_{D_{n_1}^*}\left[\left(\hat{V}_i^\pi(D_{n_1}^*) - \mathbb{E}_{D_{n_1}^*}\left[\hat{V}_i^\pi(D_{n_1}^*)\right] + \mathbb{E}_{D_{n_1}^*}\left[\hat{V}_i^\pi(D_{n_1}^*)\right] - \hat{V}_k^\pi\right)\right. \tag{24}$$

$$\left.\left(\hat{V}_j^\pi(D_{n_1}^*) - \mathbb{E}_{D_{n_1}^*}\left[\hat{V}_j^\pi(D_{n_1}^*)\right] + \mathbb{E}_{D_{n_1}^*}\left[\hat{V}_j^\pi(D_{n_1}^*)\right] - \hat{V}_k^\pi\right)\right] \tag{25}$$

$$= \mathbb{E}_{D_{n_1}^*}\left[\left(\hat{V}_i^\pi(D_{n_1}^*) - \mathbb{E}_{D_{n_1}^*}\left[\hat{V}_i^\pi(D_{n_1}^*)\right]\right)\right. \tag{26}$$

$$\left.\left(\hat{V}_j^\pi(D_{n_1}^*) - \mathbb{E}_{D_{n_1}^*}\left[\hat{V}_j^\pi(D_{n_1}^*)\right]\right)\right] \tag{27}$$

$$+ \mathbb{E}_{D_{n_1}^*}\left[\hat{V}_i^\pi(D_{n_1}^*) - \hat{V}_k^\pi\right]\mathbb{E}_{D_{n_1}^*}\left[\hat{V}_j^\pi(D_{n_1}^*) - \hat{V}_k^\pi\right] \tag{28}$$

Let $X_{n_1} := \left(\hat{V}_i^\pi(D_{n_1}^*) - \mathbb{E}_{D_{n_1}^*}\left[\hat{V}_i^\pi(D_{n_1}^*)\right]\right)$ and $Y_{n_1} := \left(\hat{V}_j^\pi(D_{n_1}^*) - \mathbb{E}_{D_{n_1}^*}\left[\hat{V}_j^\pi(D_{n_1}^*)\right]\right)$. As the empirical distribution $D_{n_1}^*$ converges to the population distribution, i.e., $D_n \xrightarrow{a.s.} D$, the resampled distribution $D_{n_1}^*$ from $D_n$ also converges to the population distribution, i.e., $D_{n_1}^* \xrightarrow{a.s.} D$. Therefore, when the estimator $\hat{V}_i^\pi(D_{n_1}^*)$ is smooth, using the continuous mapping theorem,

$$\forall i, \qquad \lim_{n_1 \to \infty} \hat{V}_i^\pi(D_{n_1}^*) = \hat{V}_i^\pi\left(\lim_{n_1 \to \infty} D_{n_1}^*\right) = \hat{V}_i^\pi(D) = c_i. \tag{29}$$

Therefore, similar to before,

$$X_{n_1}Y_{n_1} \xrightarrow{a.s.} (c_i - c_i)(c_j - c_j) = 0, \tag{30}$$

and subsequently,

$$\mathbb{E}_{D_{n_1}^*}[X_{n_1}Y_{n_1}] \xrightarrow{a.s.} 0. \tag{31}$$

Further, as $\hat{V}_k^\pi \xrightarrow{a.s.} V^\pi$,

$$\hat{V}_i^\pi(D_{n_1}^*) - \hat{V}_k^\pi \xrightarrow{a.s.} c_i - V^\pi. \tag{32}$$

Therefore,

$$\mathbb{E}_{D_{n_1}^*}\left[\hat{V}_i^\pi(D_{n_1}^*) - \hat{V}_k^\pi\right]\mathbb{E}_{D_{n_1}^*}\left[\hat{V}_j^\pi(D_{n_1}^*) - \hat{V}_k^\pi\right] \tag{33}$$

$$\xrightarrow{a.s.} (c_i - V^\pi)(c_j - V^\pi). \tag{34}$$

Using equation 31 and equation 34 in equation 28,

$$\hat{A}_{i,j} \xrightarrow{a.s.} 0 + (c_i - V^\pi)(c_j - V^\pi). \tag{35}$$

Finally, combining equation 21 and equation 35,

$$\hat{A}_{i.j} - A_{i,j} \xrightarrow{a.s.} 0. \tag{36}$$

which gives the desired result. It is worth highlighting that, theoretically, this result relies upon assumptions that the base estimators satisfy regularity conditions and are consistent. In practice, such assumptions might not hold (for e.g., when using FQE to do policy evaluation if the function approximation is under-parameterized). Nonetheless, in Section 6 we empirically illustrate that even when these assumptions are not directly satisfied, OPERA can be effective.

### A.4 Finite Sample Analysis of OPERA

Without loss of generality, let $\forall \pi \in \Pi, |J(\pi)| \leq 1$, such that we can always consider $\forall i, |\hat{V}_i^\pi| \leq 1$ (this can be trivially achieved by normalizing each estimator's output by $|V_{\max}|$). Let $V^\pi$ be a weighted sum of $\hat{V}_i^\pi$ with $\alpha^\star$, where the total number of estimators in the ensemble is $k$.

In the following, we show how the error in estimating the optimal weight coefficients $\alpha^*$ affects the MSE of the resulting estimator $\hat{\bar{V}}^\pi$. Given $\{\hat{V}_i^\pi\}_{i=1}^k$, we assume that $\hat{A}$ obtained uisng the bootstrap procedure of OPERA will produce $\hat{\alpha}$ via Equation 9 (and a resulting estimate of $\hat{\bar{V}}^\pi$). In contrast, using $A$ would have produced $\alpha^\star$ (and a resulting estimate of $\bar{V}^\pi$). To provide a finite sample characterization of OPERA's mean squared error, consider the setting where given $n$ samples in dataset $D$, there exists $\lambda > 0$, such that

$$\forall i, \quad \mathbb{E}_{D_n}[|\hat{\alpha}_i - \alpha_i^*|] \leq n^{-\lambda}, \tag{37}$$

where the expectation is over the randomness due to data $D_n$ that governs the estimates $\hat{V}_i^\pi$ and thus also the weights $\hat{\alpha}$ and $\alpha^*$ used to combines these estimates. We now provide a proof of Theorem 1. To bound the MSE of OPERA's estimate $\hat{\bar{V}}^\pi$ observe that,

$$\text{MSE}(\hat{\bar{V}}^\pi) := \mathbb{E}_{D_n}\left[\left(\hat{\bar{V}}^\pi - V^\pi\right)^2\right] \tag{38}$$

$$= \mathbb{E}_{D_n}\left[\left(\hat{\bar{V}}^\pi - \bar{V}^\pi + \bar{V}^\pi - V^\pi\right)^2\right] \tag{39}$$

$$\leq \underbrace{\mathbb{E}_{D_n}\left[(\hat{\bar{V}}^\pi - \bar{V}^\pi)^2\right]}_{=\Delta_\alpha} + \underbrace{\mathbb{E}_{D_n}\left[(\bar{V}^\pi - V^\pi)^2\right]}_{=\Delta_c} \tag{40}$$

We isolate the error of $\hat{\bar{V}}^\pi$ into two terms: $\Delta_\alpha$ and $\Delta_c$. $\Delta_c$ is the gap between the best estimate OPERA can give with $\alpha^\star$ and the true estimate of the policy performance $V^\pi$. If $V^\pi$ can be expressed as a linear combination of $\hat{V}_i^\pi$, then $\Delta_c = 0$. $\Delta_\alpha$ is the term we want to further analyze because it depends on the difference between $\hat{\alpha}$ and $\alpha^*$.

$$\Delta_\alpha := \mathbb{E}_{D_n}\left[\left(\hat{\bar{V}}^\pi - \bar{V}^\pi\right)^2\right] \tag{41}$$

$$= \mathbb{E}_{D_n}\left[\left(\sum_{i=1}^k \hat{\alpha}_i V_i^\pi - \sum_{i=1}^k \alpha_i^* V_i^\pi\right)^2\right] \tag{42}$$

$$= \mathbb{E}_{D_n}\left[\left(\sum_{i=1}^k (\hat{\alpha}_i - \alpha_i^*)\hat{V}_i^\pi\right)^2\right] \tag{43}$$

$$\leq \mathbb{E}_{D_n}\left[\left(\sum_{i=1}^k (\hat{\alpha}_i - \alpha_i^*)^2\right)\left(\sum_{i=1}^k (\hat{V}_i^\pi)^2\right)\right], \tag{44}$$

where the last inequality follows from Cauchy-Schwarz inequality. Now by using the fact that $|\hat{\theta}_i| \leq 1$ and by plugging equation 37 into equation 44:

$$\Delta_\alpha \leq \mathbb{E}_{D_n}\left[k\left(\sum_{i=1}^k (\hat{\alpha}_i - \alpha_i^*)^2\right)\right] \tag{45}$$

$$= k \sum_{i=1}^k \mathbb{E}_{D_n}\left[(\hat{\alpha}_i - \alpha_i^*)^2\right] \tag{46}$$

$$\leq \frac{k^2}{n^{2\lambda}}. \tag{47}$$

Therefore, combining equation 40 and equation 47,

$$\text{MSE}(\hat{\bar{V}}^\pi) \leq \frac{k^2}{n^{2\lambda}} + \Delta_c. \tag{48}$$

This bound factors the MSE using the term $\Delta_c$, which is the best a linear combination of estimators can do. Notice that $\Delta_c \leq \min_i \text{MSE}(\hat{V}_i^\pi)$, as the best linear combination of the estimators can at least achieve the MSE of the best estimator, by assigning weight of 1 to the best estimator and 0 to

the rest. Therefore, the rate of decay of $\Delta_c$ is bounded above by the rate of convergence of the best estimator in our ensemble.

The other term $k^2/n^{2\lambda}$ in equation 48 results due to the error in estimating $\alpha^*$ because of the bootstrapping process used for estimating $\hat{A}$ of $A$ in equation 8. This is dependent on the number of estimators $k$ – as we include more estimators in our ensemble, the combination weights $\alpha \in \mathbb{R}^k$ that need to be estimated becomes higher dimensional, thereby introducing more errors. However, the overall term decreases as the dataset size $n$ increases.

## A.5 Proofs on Properties of OPERA

### A.5.1 Invariance

In the following, we illustrate an important property of OPERA, that the resulting combined estimate $\hat{\bar{V}}^\pi$ is invariant to the addition of redundant copies of the base estimators $\{\hat{V}_i^\pi\}_{i=1}^n$. Without loss of generality, let $\hat{\mathcal{V}}_\beta \in \mathbb{R}^{(K+1)\times 1}$ be the stack of unique estimators $\{\hat{V}_i^\pi\}_{i=1}^k$ with $\hat{V}_{k+1}^\pi$ being a redundant copy of the $\hat{V}_k^\pi$,

**Theorem 3** (Invariance). *If $\hat{A}$ is positive definite, then $\hat{\bar{V}}_\beta^\pi = \hat{\bar{V}}^\pi$, where,*

$$\hat{\bar{V}}_\beta^\pi := \sum_{i=1}^{k+1} \beta_i^* \hat{V}_i^\pi \in \mathbb{R}, \qquad\qquad where, \beta^* \in \underset{\beta \in \mathbb{R}^{(k+1)\times 1}}{\arg\min} \beta^\top B\beta. \qquad (49)$$

*Proof.* We prove this by contradiction. Recall that $\hat{\alpha} \in \mathbb{R}^k$ are the weights that minimize the bootstrap estimate of MSE of $\hat{\bar{V}}^\pi$ consisting of $k$ estimators.

$$\widehat{\text{MSE}}(\hat{\alpha}_1 \hat{V}_1^\pi + ... + \hat{\alpha}_k \hat{V}_k^\pi) = \hat{\alpha}^\top \hat{A}\hat{\alpha}. \qquad (50)$$

As $\hat{V}_{k+1}^\pi$ is a redundant copy of $\hat{V}_k^\pi$,

$$\widehat{\text{MSE}}(\beta_1^* \hat{V}_1^\pi + ... + \beta_k^* \hat{V}_k^\pi + \beta_{k+1}^* \hat{V}_{k+1}^\pi) \qquad (51)$$

$$= \widehat{\text{MSE}}(\beta_1^* \hat{V}_1^\pi + ... + (\beta_k^* + \beta_{k+1}^*)\hat{V}_k^\pi) \qquad (52)$$

Finally, as $\beta^* \in \mathbb{R}^{k+1}$ is the weight that minimizes the bootstrap estimate of MSE of $\hat{\bar{V}}_\beta^\pi$. Now, if equation 50 < equation 52, then one could assign $\beta_i^* := \hat{\alpha}_i$ for $i \in \{1, ..., k\}$, and $\beta_{k+1}^* = 0$ to make equation 52 = equation 50. Further, notice that as both $\hat{\alpha}$ and $\beta^*$ are within the same feasible set of solutions, the above reassignment is also within the feasible set of solutions. Similarly, if equation 50 > equation 52, then one could assign $\hat{\alpha}_i := \beta_i^*$ for $i \in \{1, ..., k-1\}$, and $\hat{\alpha}_k = \beta_k^* + \beta_{K+1}^*$ to make equation 52 = equation 50. Hence, if equation 50 does not equal equation 52, then either $\hat{\alpha}$ or $\beta^*$ is not optimal and that would be a contradiction. This ensures that $\widehat{\text{MSE}}(\hat{\bar{V}}_\beta^\pi) = \widehat{\text{MSE}}(\hat{\bar{V}}^\pi)$.

As $\hat{A}$ is positive definite, it implies that equation 9 is strictly convex with linear constraints. Thus the minimizer $\hat{\alpha}$ of equation 9 is unique, and $\hat{\bar{V}}_\beta^\pi = \hat{\bar{V}}^\pi$. Note that due to redundancy, $B$ will not be PD despite $\hat{A}$ being PD. This would imply that there can be multiple values of $\beta_k^*$ and $\beta_{k+1}^*$. Nonetheless, since $\beta_k^* + \beta_{k+1}^* = \hat{\alpha}_k$, it implies that $\hat{\bar{V}}_\beta^\pi = \hat{\bar{V}}^\pi$.

$\square$

### A.5.2 Performance Improvement

**Theorem 4** (Performance improvement). *If $\hat{\alpha} = \alpha^*$,*

$$\forall i \in \{1, ..., k\}, \quad MSE(\hat{\bar{V}}^\pi) \le MSE(\hat{V}_i^\pi). \qquad (53)$$

*Proof.* With a slight overload of notation[2], we make the dependency of weights $\alpha$ explicit and let $\bar{V}^\pi(\alpha) = \sum_{i=1}^k \alpha_i \hat{V}_i^\pi$. Let $\text{MSE}(\bar{V}^\pi(\alpha)) := \alpha^\top A\alpha$, where $A$ is defined as in equation 2.

---

[2]Note that in Section A.4, we used $\bar{V}^\pi$ to denote $\bar{V}^\pi(\alpha^*)$. Here we make this dependence explicit.

Now from equation 1 and equation 2, we know that for $\sum_{i=1}^{k} \alpha_i = 1$,

$$\alpha^* \in \underset{\alpha \in \mathbb{R}^{k \times 1}}{\arg\min} \, \mathrm{MSE}(\bar{V}^\pi(\alpha)). \tag{54}$$

Therefore, for any $\lambda \in \mathbb{R}^{k \times 1}$ such that $\sum_{i=1}^{k} \lambda_i = 1$,

$$\mathrm{MSE}(\bar{V}^\pi(\hat{\alpha})) = \mathrm{MSE}(\bar{V}^\pi(\alpha^*)) \qquad\qquad \because \hat{\alpha} = \alpha^* \tag{55}$$

$$\leq \mathrm{MSE}(\bar{V}^\pi(\lambda)). \tag{56}$$

Notice that for $e_i \coloneqq [0, 0, .., 1, .., 0]$, where there is a $1$ in the $i^{th}$ position and zero otherwise, $\bar{V}^\pi(e_i) = \hat{V}_i^\pi$. Therefore,

$$\mathrm{MSE}(\bar{V}^\pi(\hat{\alpha})) \leq \mathrm{MSE}(\bar{V}^\pi(e_i)) \qquad\qquad \forall i \tag{57}$$

$$= \mathrm{MSE}(\hat{V}_i^\pi) \qquad\qquad \forall i. \tag{58}$$

Therefore, as $\hat{\bar{V}}^\pi = \bar{V}^\pi(\hat{\alpha})$, we have the desired result that $\forall i \in \{1, ..., k\}, \quad \mathrm{MSE}(\hat{\bar{V}}^\pi) \leq \mathrm{MSE}(\hat{V}_i^\pi)$. $\qquad\square$

### A.6 Empirical Properties of OPERA

In Figure 4: **(a)** We show that as dataset sizes increase, our bootstrap estimation of MSE approaches true MSE for each OPE estimator. **(b)** We show the MSE of OPERA with true and estimated $A$ matrix. Note both $\mathrm{MSE}(\bar{V}^\pi)$ and $\widehat{\mathrm{MSE}}(\hat{\bar{V}}^\pi)$ are near 0 and overlap each other. **(c)** We show how $\alpha$ changes between different estimators as dataset size grows.

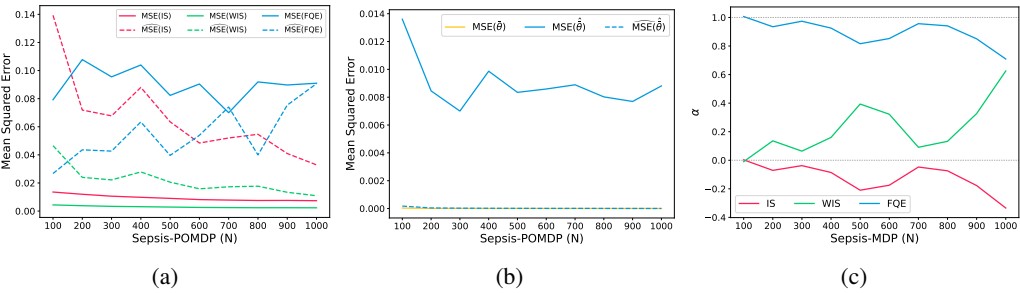

(a)            (b)            (c)

Figure 4: Properties of OPERA

### A.7 Graph Experiment

Voloshin et al. (2019) introduced a ToyGraph environment with a horizon length T and an absorbing state $x_{\mathrm{abs}} = 2T$. Rewards can be deterministic or stochastic, with +1 for odd states, -1 for even states plus one based on the penultimate state. We evaluate the considered methods on a short horizon H=4, varying stochasticity of the reward and transitions, and MDP/POMDP settings. We evaluate a single policy in this domain.

Table 5: We report the Mean-Squared Error (MSE) for the Graph domain. We conduct the experiment on Graph over 10 trials. We underscore the estimator that has the lowest MSE in the ensemble.

| Stochasticity | Observability | OPERA | BestOPE | AvgOPE | IS | WIS |
|---|---|---|---|---|---|---|
| Deterministic | MDP | **0.0339** | 0.0509 | 0.2872 | 0.7398 | 0.0509 |
| Stochastic | MDP | **0.4625** | 0.4838 | 0.7021 | 1.0803 | 0.4755 |
| Deterministic | POMDP | 1.4651 | **1.2193** | 2.3425 | 6.6273 | 0.4487 |
| Stochastic | POMDP | **0.3327** | 0.3516 | 0.3889 | 0.5634 | 0.3516 |

We report the results in Table 5. In three out of four setups, OPERA is able to produce an estimate that has a lower MSE compared to BestOPE and AvgOPE and is lower than the estimators used in the ensemble. In the deterministic POMDP setting, the IS estimate is significantly off, and OPERA

Table 6: Root Mean-Squared Error (RMSE) of the FQE estimators with different hyperparameter configurations.

| Env/Dataset | FQE 1 | FQE 2 | FQE 3 | FQE 4 |
|---|---|---|---|---|
| **Hopper** | | | | |
| medium-replay | 30.2 | 15.5 | 133.5 | 153.4 |
| medium | 52.2 | 12.5 | 242.9 | 237.6 |
| **HalfCheetah** | | | | |
| medium-replay | 126.0 | 65.0 | 439.7 | 318.8 |
| medium | 158.6 | 111.8 | 491.6 | 386.5 |
| **Walker2d** | | | | |
| medium-replay | 185.8 | 167.4 | 301.6 | 167.7 |
| medium | 184.9 | 192.0 | 406.7 | 183.8 |

is worse than BestOPE. This provides an important insight into the strengths and limitations of our particular procedure. In small tabular POMDPs, WIS can be quite good. Here, the error in the MSE estimation means that OPERA inaccurately balances the two estimators instead of relying only on WIS. We note that BestOPE does not match the performance of WIS in the ensemble. This is because the error in MSE estimation makes BestOPE erroneously choose the wrong estimator in some trials, resulting in a larger MSE. We note that on three other setups, the MSE estimation is fairly accurate. Therefore, OPERA is able to get a lower MSE than any of the OPEs in the ensemble.

These results suggest that it is important to consider the tradeoff between bootstrap estimation error and the benefit from combined estimators. When we compare the estimation quality over many policies (Sepsis and D4RL), OPERA does well, but for a particular policy, it might not outperform other estimators. This is an interesting area for future work: if and when it is possible for such a meta-algorithm to provably match the performance of any individual estimator without further assumptions.

### A.8 D4RL Experiment

**Setup** D4RL (Fu et al., 2020) is an offline RL standardized benchmark designed and commonly used to evaluate the progress of offline RL algorithms. We use 6 datasets of different quality from three environments: Hopper, HalfCheetah, and Walker2d. We choose the medium and medium-replay datasets. Medium dataset has 200k samples from a policy trained to approximately 1/3 the performance of a policy trained to completion with SAC. Medium-replay dataset takes the transitions stored in the experience replay buffer of policy – this dataset can be thought of as a dataset sampled by a mixture of policies.

**Policy Training** We train 6 policies from these three algorithms with 2 different hyperparameters for the neural network, Q-learning (CQL) (Kumar et al., 2020), implicit Q-learning (Kostrikov et al., 2021), and TD3+BC (Fujimoto et al., 2018). We initialize all neural networks (including both actor and critics, if the algorithm uses both) with the hidden dimensions of [256, 256, 256]. We train with a batch size of 512, with Adam Optimizer. We train for 100 epochs on each dataset. We only change one important hyperparameter per algorithm. We report the discounted return of each policy in Table 11,10,12. We report these scores because they are the prediction target of the FQE algorithm. We report the un-discounted return of each policy in Table 15,16,17.

| Alg | Initial $\alpha$ |
|---|---|
| CQL 1 | 1.0 |
| CQL 2 | 10 |

Table 7

| Alg | Expectile |
|---|---|
| IQL 1 | 0.7 |
| IQL 2 | 0.5 |

Table 8

| Alg | Alpha |
|---|---|
| TD3+BC 1 | 0.7 |
| TD3+BC 2 | 0.5 |

Table 9

**FQE Training** We train Fitted Q learning for each policy. As discussed in the main text, FQE has a few hyperparameter choices. We choose 4 hyperparameters for Hopper and HalfCheetah. We

| Policy | Hopper (medium-replay) | Hopper (medium) |
|---|---|---|
| CQL 1 | 193.47 | 242.24 |
| CQL 2 | 123.76 | 243.57 |
| IQL 1 | 239.20 | 246.26 |
| IQL 2 | 239.85 | 240.05 |
| TD3+BC 1 | 183.48 | 231.81 |
| TD3+BC 2 | 208.16 | 234.19 |

Table 10: Discounted perf of different policies on Hopper task.

| Policy | Walker2D (medium-replay) | Walker2D (medium) |
|---|---|---|
| CQL 1 | 252.68 | 85.39 |
| IQL 1 | 238.77 | 253.19 |
| IQL 2 | 130.29 | 243.51 |
| CQL 2 | 247.03 | 198.92 |
| TD3+BC 1 | 211.28 | 247.22 |
| TD3+BC 2 | 183.38 | 237.85 |

Table 11: Discounted perf of different policies on Walker2D task.

| Policy | HalfCheetah (medium-replay) | HalfCheetah (medium) |
|---|---|---|
| CQL 1 | 363.35 | 601.59 |
| IQL 1 | 394.06 | 436.52 |
| IQL 2 | 362.65 | 423.37 |
| CQL 2 | 354.23 | 539.03 |
| TD3+BC 1 | 407.96 | 441.20 |
| TD3+BC 2 | 318.22 | 422.65 |

Table 12: Discounted perf of different policies on HalfCheetah task.

choose another 4 hyperparameters for Walker2D. The reason is that we noticed the Q-value for Walker2D exploded if we used the same hyperparameters for the two other tasks. We should note that since OPERA does not require OPEs to be the same across tasks. The hyperparameter choices are around the Q-function neural network's hidden sizes and how many epochs we train each Q-function. Generally, training too long / over-training leads to exploding Q-values.

| Hopper/ HalfCheetah | Q-Function Network | Training Epochs |
|---|---|---|
| FQE 1 | [256, 256, 256] | 2 |
| FQE 2 | [256, 256, 256] | 3 |
| FQE 3 | [512, 512] | 1 |
| FQE 4 | [512, 512] | 2 |

Table 13: FQE Hyperparameters. Training epochs were chosen to be an early checkpoint and a late checkpoint (before exploding Q-values).

| Walker2D | Q-Function Network | Training Epochs |
|---|---|---|
| FQE 1 | [128, 256, 512] | 2 |
| FQE 2 | [128, 256, 512] | 5 |
| FQE 3 | [512, 512] | 1 |
| FQE 4 | [512, 512] | 2 |

Table 14: FQE Hyperparameters. Training epochs were chosen to be an early checkpoint and a late checkpoint (before exploding Q-values).

## A.9 Sepsis and Graph Experiment Details

### A.9.1 Sepsis

The first domain is based on the simulator and works by Oberst and Sontag (2019) and revolves around treating sepsis patients. The goal of the policy for this simulator is to discharge patients from the hospital. There are three treatments the policy can choose from antibiotics, vasopressors, and mechanical ventilation. The policy can choose multiple treatments at the same time or no treatment at all, creating 8 different unique actions.

The simulator models patients as a combination of four vital signs: heart rate, blood pressure, oxygen concentration and glucose levels, all with discrete states (for example, for heart rate low, normal and high). There is a latent variable called diabetes that is present with a $20\%$ probability which drives the likelihood of fluctuating glucose levels. When a patient has at least 3 of the vital signs simultaneously

| Policy | Hopper (medium-replay-v2) | Hopper (medium-v2) |
|---|---|---|
| CQL 1 | 433.40 | 2550.03 |
| CQL 2 | 439.56 | 2787.95 |
| IQL 1 | 3144.02 | 1768.19 |
| IQL 2 | 2177.90 | 2028.27 |
| TD3 1 | 1104.04 | 1977.88 |
| TD3 2 | 910.26 | 1751.87 |

Table 15: Undiscounted perf of different policies on Hopper task.

| Policy | Walker2D (medium-replay-v2) | Walker2D (medium-v2) |
|---|---|---|
| CQL 1 | 3732.01 | 145.25 |
| IQL 1 | 2383.09 | 3044.03 |
| IQL 2 | 776.79 | 3194.87 |
| CQL 2 | 3073.49 | 1409.01 |
| TD3 1 | 2250.07 | 3920.79 |
| TD3 2 | 1656.82 | 3732.23 |

Table 16: Undiscounted perf of different policies on Walker2D task.

| Policy | HalfCheetah (medium-replay-v2) | HalfCheetah (medium-v2) |
|---|---|---|
| CQL 1 | 4053.04 | 7894.69 |
| CQL 2 | 4192.01 | 6875.66 |
| IQL 1 | 4995.02 | 5704.12 |
| IQL 2 | 4657.00 | 5475.88 |
| TD3 1 | 5324.46 | 5758.83 |
| TD3 2 | 5002.90 | 5420.27 |

Table 17: Undiscounted perf of different policies on HalfCheetah task.

out of the normal range, the patient dies. If all vital signs are within normal ranges and the treatments are all stopped, the patient is discharged. The reward function is $+1$ if a patient is discharged, $-1$ if a patient dies, and 0 otherwise. We truncate the trajectory to 20 actions (H=20). For this simulator, early termination means we don't get to observe a positive or negative return on the patient.

We follow the process described by Oberst and Sontag (2019) to marginalize an optimal policy's action over 2 states: glucose level and whether the patient has diabetes. This creates the Sepsis-POMDP environment. We sample 200 and 1000 patients (trajectories) from Sepsis-POMDP environment with the optimal policy that has 5% chance of taking a random action. We also sample trajectories from the original MDP using the same policy; we call this the Sepsis-MDP environment.

**FQE Training** We use tabular FQE. Therefore, there is no representation mismatch. We additionally use cross-fitting, a form of procedure commonly used in causal inference (Chernozhukov et al., 2016). Cross-fitting is a sample-splitting procedure where we swap the roles of main and auxiliary samples to obtain multiple estimates and then average the results. The main goal of cross-fitting is to reduce overfitting. We notice significant performance improvement of our FQE estimator after using cross-fitting. We present the RMSE of each of our trained FQE estimator in Table 6.

### A.9.2 Graph

For the graph environment, we set the horizon H=4, with either POMDP or MDP and ablate on the stochasticity of transition and reward function. The optimal policy for the Graph domain is simply the policy that chooses action 0. All the experiments reported have 512 trajectories.

### A.10 Compute Resource

The training was done on a small cluster of 6 servers, each with 16GB RAM, and 4-8 GPUs of Nvidia A5000. D4RL was the most computationally expensive experiment.

