# OpenReview forum: "OPERA: Automatic Offline Policy Evaluation with Re-weighted Aggregates of Multiple Estimators"
_NeurIPS.cc/2024/Conference — NeurIPS 2024 poster_

### Official Review · Reviewer_6uiB · 2024-06-25

**Soundness:** 4
**Presentation:** 3
**Contribution:** 3
**Rating:** 6
**Confidence:** 3

**Summary:**

The paper addresses the challenge of evaluating new sequential decision-making policies using OPE techniques. The authors propose a new algorithm, OPERA, which adaptively blends multiple OPE estimators to improve the accuracy of policy performance estimates without relying on explicit selection methods. This approach leverages bootstrapping to estimate the mean squared error of different estimator weightings and optimizes it as a convex problem. The proposed method is shown to be consistent and demonstrates superior performance in selecting higher-performing policies in healthcare and robotics compared to existing approaches. This contributes to a more general-purpose, estimator-agnostic framework for offline reinforcement learning.

**Strengths:**

The paper formulates an interesting problem of how to aggregate different OPE estimators to achieve better(lower) MSE in a data-driven way


The paper provides interesting empirical investigations regarding the tuned alpha in Figure 1, which is interesting and helpful to understand how the method works


The paper provides promising empirical results on both contextual bandit and reinforcement learning environments, showing that, by coming multiple estimators, we can achieve better estimations in a range of situations

**Weaknesses:**

There still remains an important question of how to construct an appropriate set of estimators before performing the proposed ensemble algorithm


I still do not get an intuition of if the proposed method solves fundamentally an easier problem than standard OPE. I mean, if we know the MSE, then it is really straightforward that the proposed algorithm works well, but we need to estimate it somehow, which includes the true policy value, which is the estimand of OPE. Therefore, intuitively, MSE estimation has the same difficulty as OPE, but the proposed algorithm works better than OPE. I’d like to know in what sense the proposed algorithm makes OPE easier.


In theorem 1, it seems that the error rate of \alpha estimation is given, which is not guaranteed, if my understanding is correct.

**Questions:**

How should we construct the set of estimators to perform the proposed algorithm? Can the authors propose any general guideline?


What would be the intuition of the better effectiveness of the proposed method against standard OPE? Some reasonable accuracy of the proposed method means that we can estimate the MSE somewhat accurately, meaning that we can do OPE similarly accurately, in my intuition.


In theorem 1, the error rate of \alpha estimation is not guaranteed, right?


When combining different estimators, can we consider more complex functions such as polynomials or more general function classes rather than just linear combinations as done in the paper?

**Limitations:**

Yes

---

> ### Author Rebuttal · Authors · 2024-08-07
>
> We thank the reviewer for their thoughtful review.
>
> > How should we construct the set of estimators to perform the proposed algorithm? Can the authors propose any general guideline?
>
> Thank you for the suggestion. We are adding guidelines as a section in the appendix, and we summarize them here:
> 1. The estimators should be selected based on the properties of the domain well. For example, if the task horizon is short, then IS estimators would be great, but if the task horizon is long, we should not include IS estimators.
> 2. If there are estimators known to under or over-estimate the policy performance, then selecting a balanced set of such estimators can allow OPERA to cancel out the bias of these estimators (see the discussion in Sec 4.1 Interpretability).
> 3. Choosing a reasonable number of estimators (starting with K=3) and only include more when the dataset size grows.
>
> >  What would be the intuition of the better effectiveness of the proposed method against standard OPE?
>
> Thank you for the good question. The reviewer is correct that OPERA uses estimates of the MSE, which can seem intuitively to be the same hardness as OPE estimation. However, OPERA fundamentally is an ensemble method (similar to stacking in statistics) that combines multiple estimators. For OPERA to offer an improvement over some of the input OPE estimators, we do not have to have a perfect estimate of their MSE, but a good enough estimate that we can combine across them. Note that OPERA with an estimated MSE per estimator is not guaranteed to be as good as picking the (unknown) estimator with the true best MSE. However, this estimator is unknown because, as the reviewer notes, we do not know the true MSE.  We find empirically that the bootstrap estimates provide a reasonable enough approximation that we can find a decent weighting using OPERA that yields an estimator that often (see e.g. Table 1) improves over all individual input estimators, as Figure 1 (and the last part of Section 4) discusses should be possible in some cases given the true (unknown) MSE. We are happy to add additional discussion around this point in the paper. We also note that we think further work on better MSE estimation could further improve OPERA.
>
> > In theorem 1, the error rate of \alpha estimation is not guaranteed, right?
>
> The reviewer is correct, and we will make sure this is clear. The MSE of OPERA has a factor of $\lambda$.
>
> > When combining different estimators, can we consider more complex functions such as polynomials or more general function classes rather than just linear combinations as done in the paper?
>
> Yes, we completely agree this would be an interesting direction for future work. We choose to focus initially on a linearly weighted estimator, which allows us to decompose OPERA’s MSE as a function of the underlying estimators’ MSE: see the derivation in Remark 1. This also allowed the optimization objective to be a convex quadratic program. Other choices of combination might result in different non-convex optimization objectives which, depending on the solution techniques, might introduce additional approximations in OPERA’s policy value estimate. Still, we agree more complicated functions would be worth investigating since they give additional modeling flexibility.
>
> Did this answer your questions? We are also happy to answer more questions if they arise.

---

> > ### Comment · Reviewer_6uiB · 2024-08-13
> >
> > Thank you to the authors for the useful clarification. I will maintain my positive assessment.

---

### Official Review · Reviewer_BP3B · 2024-07-04

**Soundness:** 4
**Presentation:** 4
**Contribution:** 3
**Rating:** 8
**Confidence:** 3

**Summary:**

The paper "OPERA: Automatic Offline Policy Evaluation with Re-weighted Aggregates of Multiple Estimators" deals with the challenge of evaluating new decision-making policies using past data, which is vital in areas like healthcare and education where mistakes can be costly.
The authors introduce OPERA, a new algorithm that improves the accuracy of policy evaluation by combining several existing estimators instead of relying on just one. This approach makes the evaluations more reliable.
They provide a solid theoretical basis for OPERA, showing that it effectively minimizes errors and meets key evaluation criteria. A standout feature is using bootstrapping to estimate the mean squared error (MSE) of each estimator, which helps in finding the best way to combine them.
The paper demonstrates OPERA's effectiveness through tests in various fields, such as healthcare and robotics. It consistently outperforms other methods by selecting better policies and achieving lower errors.
One of the key contributions is OPERA's flexibility. It can be used in many applications without needing strict conditions on the input estimators, making it a versatile tool for different offline reinforcement learning tasks.
In summary, the paper presents a new method for policy evaluation that combines multiple estimators to enhance accuracy and reliability, backed by strong theoretical and practical evidence.

**Strengths:**

Originality:
This paper brings an innovative approach to offline policy evaluation with the OPERA algorithm. Instead of relying on a single estimator, OPERA cleverly combines multiple estimators to improve accuracy. This innovation taps into the strengths of different estimators, solving a major limitation of existing methods. The creative use of bootstrapping to estimate mean squared error (MSE) adds to the algorithm's robustness, making it a standout contribution.
Quality:
The quality of this paper is excellent. The authors do a fantastic job of defining the problem of offline policy evaluation and explaining why current methods fall short. They clearly describe their proposed solution, the OPERA algorithm, and walk the reader through each step in a logical and detailed manner.
The theoretical groundwork is solid. The authors back up their claims with rigorous proofs, showing that OPERA effectively minimizes mean squared error (MSE) and meets important criteria for policy evaluation. They also anticipate potential issues with existing methods, like bias and variance, and introduce innovative solutions to tackle these challenges using bootstrapping.
The empirical validation is thorough and convincing. The authors test OPERA across various fields, including healthcare (like sepsis treatment) and robotics, demonstrating its effectiveness. They detail their experimental setup, the datasets used, and the baseline methods for comparison. The results are clear: OPERA consistently outperforms other methods by selecting better policies and achieving lower errors, which strongly supports the practical value of their algorithm.
Clarity:
The paper is well-structured and clearly written. The flow is logical and easy to follow, from problem definition to solution proposal and validation.
Significance:
This paper makes a significant impact in offline reinforcement learning and policy evaluation. By combining multiple estimators, the authors solve a crucial problem, especially in critical fields like healthcare and education. The results show that OPERA consistently outperforms existing methods, proving its value for both researchers and practical applications.

**Weaknesses:**

While the theoretical and empirical aspects are well-covered, the paper could really use a more detailed discussion on how to implement OPERA in practice. Offering guidelines or best practices for using OPERA in different situations would be very helpful for practitioners.

**Questions:**

The paper primarily compares OPERA with traditional methods. Have you considered comparing OPERA with more recent state-of-the-art models? Including such comparisons could provide a clearer picture of OPERA's competitive edge.

**Limitations:**

The authors have done a great job addressing the limitations of their work. They recognize the challenges of combining multiple estimators and provide a solid theoretical foundation to back up their approach. They also discuss using bootstrapping to tackle issues like bias and variance, which is a key part of their method.
However, it would be helpful to add a separate section that dives into potential edge cases and limitations in more detail. This could cover situations where OPERA might have difficulties, such as dealing with very noisy or incomplete data, and practical challenges users might face during implementation.

---

> ### Author Rebuttal · Authors · 2024-08-07
>
> We thank the reviewer for the thoughtful review.
>
> > the paper could really use a more detailed discussion on how to implement OPERA in practice. Offering guidelines or best practices for using OPERA in different situations would be very helpful for practitioners.
>
> Thank you for the suggestion. We are adding guidelines as a section in the appendix, and we summarize them here:
> 1. The estimators should be selected based on the properties of the domain well. For example, if the task horizon is short, then IS estimators would be great; but if the task horizon is long, we should not include IS estimators.
> 2. If there are estimators known to under or over-estimate the policy performance, then selecting a balanced set of such estimators can allow OPERA to cancel out the bias of these estimators (see the discussion in Sec 4.1 Interpretability).
> 3. Choosing a reasonable number of estimators (starting with K=3) and only include more when the dataset size grows.
>
> > The paper primarily compares OPERA with traditional methods. Have you considered comparing OPERA with more recent state-of-the-art models?
>
> Thank you for the suggestion. We do compare with methods such as SLOPE [1] and BVFT [2] which to our knowledge are state of the art. On the contextual bandit domain, we show that OPERA outperforms SLOPE across 180 conditions when the dataset size is larger than 300. For more realistic robotic control tasks (D4RL), OPERA outperforms BVFT on three tasks: Hopper, HalfCheetah, and Walker2d. If there are other additional algorithms the reviewer was thinking of, please let us know as we’d be happy to look into these.
>
> [1] Su, Y., Srinath, P., and Krishnamurthy, A. (2020). Adaptive estimator selection for off-policy evaluation. In International Conference on Machine Learning, pages 9196–9205. PMLR.
>
> [2] Xie, T. and Jiang, N. (2021). Batch value-function approximation with only realizability. In International Conference on Machine Learning, pages 11404–11413. PMLR.
>
> Thank you for the review. Did this answer your questions? We are also happy to answer more questions if they arise.

---

> > ### Comment · Reviewer_BP3B · 2024-08-07
> >
> > I have read the Rebuttal, thanks for answering my question.

---

### Official Review · Reviewer_QbuQ · 2024-07-07

**Soundness:** 3
**Presentation:** 4
**Contribution:** 3
**Rating:** 7
**Confidence:** 4

**Summary:**

The authors propose a novel offline policy evaluation algorithm, that linearly blends the estimates from many OPE estimators to produce a combined estimate that achieves a lower MSE.

**Strengths:**

The paper is very well written, complete, and easy to read.

The experiments are well executed, the methods are evaluated in numerous different domains and compared against other relevant baselines in each of them.

I specifically like the discussion on different MSE estimation strategies in Section 6, which shows that the authors really thought about the problem and various choices made in their method.

**Weaknesses:**

The authors propose to estimate MSE of \hat{V}_i by estimating \pi using bootstrapped D_n and calculating the squared error from an estimate of \pi done by \hat{V}_i using the original D_n dataset. I think this underestimates bias. For example, if \hat{V}_i returns a constant for any dataset and policy, then its MSE would be 0, and I assume \hat{\alpha}_i would be 1, meaning the ensemble reward would correspond to this constant. I appreciate the variants of OPERA presented in 6.2 that partially address this.

Minor:
* [L114] D_n is defined both as 1 to n and 0 to n
* [L125] \theta_* is undefined
* [L136] Eq. 4 is referred although it is not labeled when defined (and many others)
* [L169] \theta_* is still undefined
* [L189] The word equation is repeated twice, I recommend using the LaTeX package cleveref

**Questions:**

Considering the first paragraph in weaknesses, can you discuss whether OPERA systematically favors biased estimators?

Recent work of Cief et al. (2024) provides a new way of estimating MSE, which can potentially improve OPERA.  As this was published after the NeurIPS deadline, I do not consider it an issue.

Cief, Matej, Michal Kompan, and Branislav Kveton. “Cross-Validated Off-Policy Evaluation.” arXiv, May 24, 2024. http://arxiv.org/abs/2405.15332.

**Limitations:**

The authors provide a thorough discussion on the limitations in Section 6 and Appendix A.7.

---

> ### Author Rebuttal · Authors · 2024-08-07
>
> We thank the reviewer for the thoughtful review and pointing out minor errors. We have corrected the inconsistent symbols and clerical mistakes. Much appreciated!
>
> > The authors propose to estimate MSE of \hat{V}_i by estimating \pi using bootstrapped D_n and calculating the squared error from an estimate of \pi done by \hat{V}_i using the original D_n dataset. I think this underestimates bias. For example, if \hat{V}_i returns a constant for any dataset and policy, then its MSE would be 0… I appreciate the variants of OPERA presented in 6.2 that partially address this.
>
> > …Considering the first paragraph in weaknesses, can you discuss whether OPERA systematically favors biased estimators?
>
> Thank you for asking about this. We don’t believe that OPERA systematically favors biased estimators— for example, if there is an estimator which always returns -10^6 or 10^6 with 50/50 probability, it would have a high variance (in addition to being biased) and OPERA would likely place low weight on this estimator. We do completely agree that the method (prior to 6.2) can underestimate the bias. Indeed, using Bootstrap to estimate MSE reliably assumes that the estimator will approach the true target asymptomatically. Therefore, an estimator that only produces a constant value invalidates the Bootstrap procedure, and its MSE cannot be reliably estimated with Bootstrap. One way to address this is by using estimators that have known convergence guarantees.
>
> We agree that empirically estimators that have very low variance but are potentially very biased (i.e., model-based estimators or FQEs) might be favored by OPERA. As the reviewer notes, this helped motivate the modifications we present in Section 6.2, where we use a consistent estimator as the centering variable/estimator.
>
> We will expand our discussion of these issues in the paper.
>
> > Recent work of Cief et al. (2024) provides a new way of estimating MSE, which can potentially improve OPERA. As this was published after the NeurIPS deadline, I do not consider it an issue.
>
> Thank you for suggesting this paper– we look forward to going through it in more detail, though, as the reviewer notes, it was released after the deadline.
>
> Thank you for the review. Did this answer your questions? We are also happy to answer more questions if they arise.

---

> > ### Comment · Reviewer_QbuQ · 2024-08-07
> >
> > Thank you for answering my question. After reviewing other reviews and their discussions, I decided to keep my score and vote for the paper's acceptance.

---

### Official Review · Reviewer_Rbow · 2024-07-12

**Soundness:** 3
**Presentation:** 4
**Contribution:** 2
**Rating:** 6
**Confidence:** 4

**Summary:**

The paper introduces OPERA, an algorithm for offline policy evaluation (OPE) in reinforcement learning (RL). OPERA addresses the challenge of selecting the best OPE estimator by adaptively combining multiple estimators using a statistical procedure that optimizes their weights to minimize mean squared error (MSE). The authors prove OPERA's theoretical consistency and desirable properties, ensuring it is at least as accurate as any individual estimator. The algorithm employs a bootstrapping method to estimate MSE, circumventing the need for direct access to true performance measures.

**Strengths:**

1. The paper presents a novel approach to offline policy evaluation by introducing OPERA, a method that adaptively blends multiple OPE estimators using a statistically optimized weighting mechanism. This approach is innovative as it leverages the strengths of various estimators without requiring explicit selection. The use of bootstrapping to estimate mean squared error (MSE) and optimize the estimator weights is a simple yet creative combination of existing statistical techniques in a new context.

2. The paper is well-written and clearly structured. The problem statement, methodology, theoretical analysis, and experimental results are presented in a logical and coherent manner. The use of mathematical formulations and proofs is precise, aiding in the clear communication of complex concepts. Additionally, the inclusion of diagrams and pseudocode for the OPERA algorithm enhances understanding and provides readers with a clear roadmap of the proposed method.

**Weaknesses:**

**Assumption of Consistent Estimators**
The theoretical guarantees provided for OPERA rely on the assumption that at least one of the base estimators is consistent. However, in practice, this assumption may not always hold, especially in complex or noisy environments where all available estimators could be biased or inconsistent. The paper could benefit from a discussion on how OPERA performs under such conditions and whether there are ways to relax this assumption while still maintaining acceptable performance. For example, could the authors design an experiment where the state, action and/or reward are severely imbalanced, which leads to insufficient data coverage and approximation error to behaviour policy? A practical reflection can be found in a critical study of real-world ICU treatment [1].

[1] Luo, Zhiyao, et al. "Position: Reinforcement Learning in Dynamic Treatment Regimes Needs Critical Reexamination." Forty-first International Conference on Machine Learning.

**Questions:**

1.The theoretical guarantees of OPERA rely on the assumption that at least one base estimator is consistent. How does OPERA perform when this assumption does not hold? Are there any mechanisms within OPERA to detect and handle inconsistent base estimators?

2. What practical considerations should be taken into account when implementing OPERA in real-world scenarios? Are there specific guidelines for choosing the initial set of estimators or criteria for including new estimators in the ensemble?

**Limitations:**

This paper seems to lack a section for limitation. I recommend adding 1 short paragraph to the conclusion section to summarize the border limitations and social impact.

---

> ### Author Rebuttal · Authors · 2024-08-07
>
> We thank the reviewer for their thoughtful feedback! We incorporated these discussions into the paper but respond to them individually here.
>
> > OPERA rely on the assumption that at least one of the base estimators is consistent. However, in practice, this assumption may not always hold. The paper could benefit from a discussion on how OPERA performs under such conditions and whether there are ways to relax this assumption while still maintaining acceptable performance. For example, could the authors design an experiment where the state, action and/or reward are severely imbalanced, which leads to insufficient data coverage.
>
> > How does OPERA perform when this assumption does not hold? Are there any mechanisms within OPERA to detect and handle inconsistent base estimators?
>
> Thanks for raising this important issue. In general, if there is good coverage over states and actions, then including an IS estimator is sufficient to ensure that a consistent base estimator exists. However, if there is not good coverage, that will result in poor inconsistent estimators. OPERA is fundamentally an ensemble method (similar to stacking in statistics) that combines multiple estimators. As long as we can accurately estimate the MSE of the estimators in the reward/action imbalanced conditions, then OPERA will be able to improve upon base estimates.
>
> As for mechanisms, we can introduce ideas such as KL divergence between the behavior/sampling policy and the evaluation policy, which will indicate when state/action/reward imbalance might occur. If this is the case, most OPE methods, in general, will have issues returning an accurate estimate. We would recommend collecting additional data before the evaluation or including estimators that can produce reliable OPE estimates in this case.
>
> > What practical considerations should be taken into account when implementing OPERA in real-world scenarios? Are there specific guidelines for choosing the initial set of estimators or criteria for including new estimators in the ensemble?
>
> Thank you for this question! We are adding guidelines as a section in the appendix, and we summarize them here:
> 1. The estimators should be selected based on the properties of the domain well. For example, if the task horizon is short, then IS estimators are often useful to include.
> 2. If there are estimators known to under or over-estimate the policy performance, then selecting a balanced set of such estimators can allow OPERA to cancel out the bias of these estimators (see the discussion in Sec 4.1 Interpretability).
> 3. Choosing a reasonable number of estimators (starting with K=3) and only include more when the dataset size grows.
>
> > This paper seems to lack a section for limitation. I recommend adding 1 short paragraph to the conclusion section to summarize the border limitations and social impact.
>
> We have added text in the conclusion to address the limitation. Thank you for the suggestion!
>
> Thank you for the review. Did this answer your questions? We are also happy to answer more questions if they arise.

---

### Decision · Program_Chairs · 2024-09-25

**Decision:**

Accept (poster)

**Comment:**

This paper presents OPERA, a novel and estimator-agnostic approach to offline policy evaluation that significantly improves accuracy by adaptively blending multiple OPE estimators. The innovative use of bootstrapping to estimate the mean squared error of different estimator weightings, and its subsequent optimization as a convex problem, demonstrates a sound theoretical foundation. Comprehensive empirical evaluation across domains like healthcare and robotics showcases OPERA's superior performance in selecting higher-performing policies compared to existing approaches. This work represents a valuable contribution to the field of offline reinforcement learning, and thus I recommend acceptance of this paper.